# Development of innovative multi-epitope mRNA vaccine against central nervous system tuberculosis using in silico approaches

Huidong Shi[1], Yuejie Zhu[2], Kaiyu Shang[1], Tingting Tian[1], Zhengwei Yin[1], Juan Shi[1], Yueyue He[3], Jianbing Ding[1], Quan Wang[4]*, Fengbo Zhang📧[1]*

1 State Key Laboratory of Pathogenesis, Prevention and Treatment of High Incidence Diseases in Central Asia, The First Affiliated Hospital of Xinjiang Medical University, Urumqi, China, 2 Reproductive Medicine Center, The First Affiliated Hospital of Xinjiang Medical University, Urumqi, China, 3 Department of Immunology, School of Basic Medical Sciences, Xinjiang Medical University, Urumqi, China, 4 Department of Clinical Laboratory, The Eighth Affiliated Hospital of Xinjiang Medical University, Urumqi, China

* 3197606@qq.com (QW); fengbozhang@xjmu.edu.cn (FZ)

## Abstract

Tuberculosis(TB) of the Central nervous system (CNS) is a rare and highly destructive disease. The emergence of drug resistance has increased treatment difficulty, leaving the Bacillus Calmette-Guérin (BCG) vaccine as the only licensed preventative immunization available. This study focused on identifying the epitopes of *PknD* (*Rv0931c*) and *Rv0986* from *Mycobacterium tuberculosis*(*Mtb*) strain H37Rv using an in silico method. The goal was to develop a therapeutic mRNA vaccine for preventing CNS TB. The vaccine was designed to be non-allergenic, non-toxic, and highly antigenic. Codon optimization was performed to ensure effective translation in the human host. Additionally, the secondary and tertiary structures of the vaccine were predicted, and molecular docking with TLR-4 was carried out. A molecular dynamics simulation confirmed the stability of the complex. The results indicate that the vaccine structure shows effectiveness. Overall, the constructed vaccine exhibits ideal physicochemical properties, immune response, and stability, laying a theoretical foundation for future laboratory experiments.

## 1. Introduction

Primarily affecting humans, *Mtb* is an acid-fast aerobic, non-motile, spore-forming bacterium [1]. According to the World Health Organization (WHO) Global TB Report 2022,10.6 million people were diagnosed and 1.6 million people died from TB in 2021. This marks a 3.6% increase from 2020 [2]. It is worth noting that CNS TB is a form of TB, predominantly presenting as tuberculous meningitis (TBM) with a notably high mortality rate [3]. In patients with HIV, the mortality rate for TBM is close to 50% [4]. At the population level, the incidence is highest in children aged 2–4 years [5]. Previous studies have shown that *Mtb* deposits develop during hematogenous dissemination in the brain parenchyma and meninges, leading to the gradual formation of tuberculoma. The physical rupture of the tuberculoma allows the bacterium to spread directly into the cerebrospinal fluid (CSF), ultimately resulting in tuberculous meningitis. Clinical symptoms primarily manifest as infarction due to vasculitis [6]. Children

**Funding:** This study was supported by State Key Laboratory of Pathogenesis, Prevention and Treatment of High Incidence Diseases in Central Asia (SKL-HIDCA2021-JH11, https://caskl.xjmu.edu.cn/); National Natural Science Foundation of China Regional Science Foundation Project, 82360394,https://www.nsfc.gov.cn/publish/portal0/tab442/info92109.htm; Youth Science and technology top talent Program(2022TSYCCX0112, http://kjt.xinjiang.gov.cn/kjt/c100870/201111/acd5ead987f945cba2498ce413a36388.shtml); Outstanding Youth Science Foundation of Xinjiang Uygur Autonomous Region(2023D01E12,http://kjt.xinjiang.gov.cn/kjt/index.shtml) and Xinjiang Uygur Autonomous Region science and technology support project (2022E02061,http://kjt.xinjiang.gov.cn/kjt/c100870/201111/acd5ead987f945cba2498ce413a36388.shtml) Author: F.B Zhang. The funders had no role in study design, data collection and analysis, decision to publish, or preparation of the manuscript.

**Competing interests:** The authors have declared that no competing interests exist.

**Abbreviations:** BCG, Bacillus Calmette-Guérin; CAI, Codon adaptation index; CBE, Conformational B-cell epitopes; CNS, Central nervous system; CTL, Cytotoxic T Lymphocyte; GRAVY, Grand average of hydropathicity; HLA, Human leukocyte antigen; HTL, Helper T Lymphocyte; IEDB, Immune Epitope Database; LBE, Linear B-cell epitopes; MEV, Multiple epitope vaccine; MFE, Minimal free energy; MHC, Major histocompatibility complex; MITD, MHC I-targeting domain; Mtb, Mycobacterium tuberculosis; PCR, Polymerase chain reaction; RMSD, Root mean square deviations; RMSF, Root-mean-square fluctuation; TB, Tuberculosis; TBM, tuberculous meningitis; TLR4, Toll-like receptor 4; UTR, Untranslated Region.

and HIV co-infected individuals are considered high-risk groups for CNS TB [7]. Despite the existence of antibiotics and effective treatment options, there are several chemotherapy-related complications that persist, such as prolonged treatment durations, severe adverse reactions, poor adherence, and the emergence of multidrug resistance. Consequently, the global management of anti-TB treatment encounters significant obstacles [8–10].

Therapeutic vaccination has emerged as a potential new approach for treating TB [11]. While prophylactic vaccines like BCG can help prevent *Mtb* infection or active TB development, therapeutic vaccines aim to prevent recurrence post-cure or serve as adjuvant therapy [12]. Current TB vaccine candidates include inactivated, live attenuated, subunit, viral vector, DNA, and mRNA vaccines [13]. The large-scale production of mRNA vaccines has shown a trend towards industrialization during the COVID-19 epidemic [14]. However, no mRNA vaccine has yet been developed for CNS TB. mRNA vaccines work by transferring exogenous mRNA encoding antigen into cells through the expression system, leading to an immune response upon antigen synthesis [15, 16]. These vaccines offer several advantages. Firstly, mRNA can encode and express all genetic information of various proteins, allowing for optimized vaccine development through mRNA sequence modification [17]. Secondly, most mRNA vaccines can be produced and purified using similar procedures, which paves the way for the development of other similar mRNA vaccines [15]. Lastly, in vitro transcription simplifies the production of mRNA vaccines [17].

In this study, two proteins, *PknD* and *Rv0986*, from *Mtb* strain H37Rv were examined. *PknD* (*Rv0931c*) encodes a eukaryotic serine-threonine protein kinase with extracellular (sensor) and intracellular kinase domains that are uniquely found in CNS TB [18]. On the other hand, *Rv0986* is a component of the ABC transporter complex and plays a role in host cell binding through secretion of adhesion factors or maintenance of mycobacterium cell envelope integrity, also specific to CNS TB [19]. The objective of this research was to develop a new multi-epitope mRNA therapeutic vaccine targeting a protein associated with CNS TB. Various in silico methods which had been used in previous studies such as Immune Epitope Database (IEDB), NetCTLpan1.1, NetMHCIIpan-4.0, and SVMtrip were employed to analyze the antigen epitopes of *PknD* and *Rv0986*. Additionally, Cytotoxic T lymphocyte(CTL), Helper T lymphocyte(HTL), and B cell epitopes were connected using AAY, GPGPG, and KK linkers [20]. To investigate the relationship between T cells and alleles, molecular docking was conducted [21]. Furthermore, an analysis of the physicochemical properties, antigenicity, allergenicity, and toxicity of the mRNA vaccine was performed, followed by immune simulations to validate the hypothesis. Codon optimization of the mRNA vaccine was carried out to ensure accurate translation in the host. Subsequently, predictions on the secondary and tertiary structures of the vaccine were made. The resulting tertiary structure was then molecularly docked with TLR-4. Finally, a molecular dynamics simulation was utilized to confirm the stability of the compound [22].

## 2. Materials and methods

### 2.1 Sequence source

The amino acid sequences of the target proteins *PknD* and *Rv0986* were retrieved from the UniProt database(https://www.uniprot.org/). Homology analysis of these selected proteins was conducted using MAFFT(version 7) and showed in Jalview(2.11.3.3) software [23]. The research process of this paper is shown in (Fig 1).

### 2.2 Selection of target proteins

The software Prot Param(https://web.expasy.org/protparam/) was used to analyze the physico-chemical properties of the selected proteins and MEVs. The antigenicity of the protein was

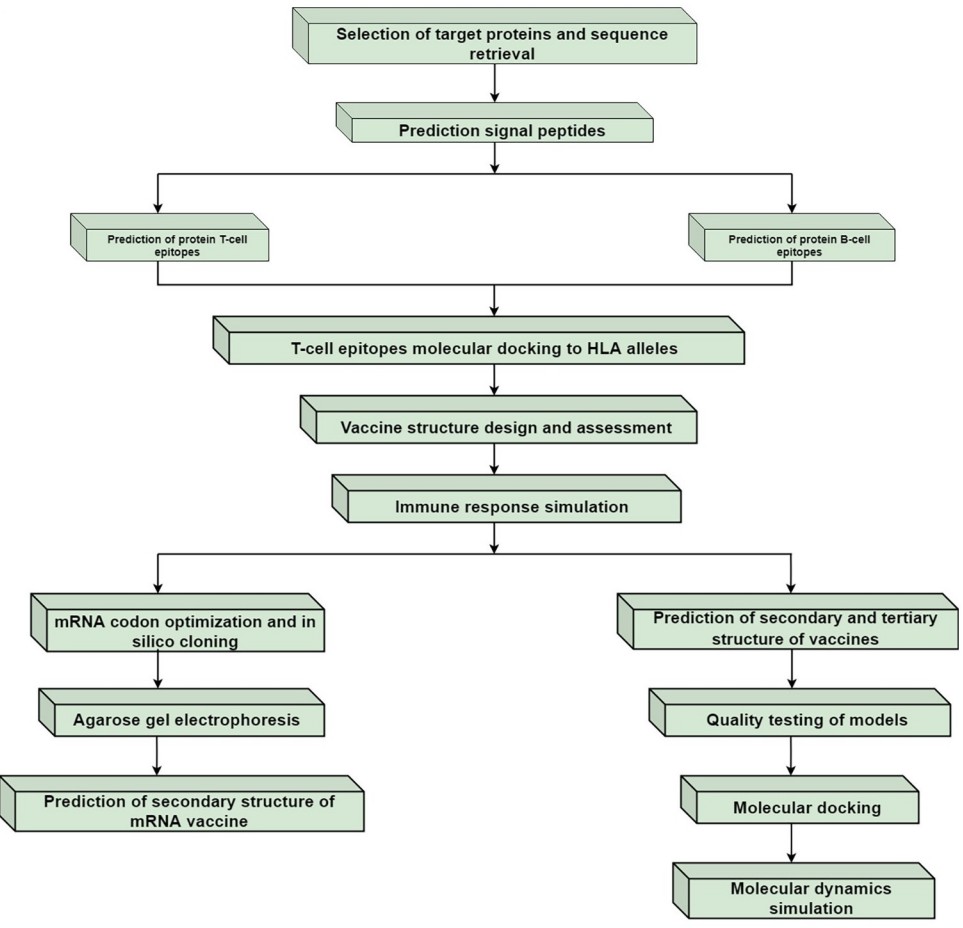

**Fig 1. The experimental process of this study.**

analyzed using Vaxi Jen2.0(http://www.ddg-pharmfac.net/vaxijen/VaxiJen/VaxiJen.html) [24]. AllergenFPv.1.0(http://www.ddg-pharmfac.net/AllerTOP)was used to analyze the sensitization and ToxinPred(https://webs.iiitd.edu.in/raghava/toxinpred/multi_submit.php) was used to analyze the toxicity of the protein.

## 2.3 Prediction of signal peptides

Before predicting protein epitope, the signal peptide sequence should be removed first. We used SignalP6.0(https://services.healthtech.dtu.dk/service.php?SignalP-6.0) [25] and Signal BLAST(http://sigpep.services.came.sbg.ac.at/signalblast.html) [26] to predict the signal peptide of the protein.

## 2.4 Prediction of protein T-cell epitopes

Major histocompatibility complex (MHC) molecules play a crucial role in binding and presenting antigenic peptides for recognition by T lymphocytes. MHC class I molecules interact with the CD8T cell subset, while MHC class II molecules interact with the CD4T cell subset. Human leukocyte antigen (HLA) genes are synonymous with human MHC genes [22, 27]. HLA alleles exhibit specificity based on geographical and population factors [28]. In this study,

we focused on alleles commonly found in Xinjiang (HLA-A*1101, HLA-A*0201, HLA-A*0301, HLA-DRB1*0701, HLA-DRB1*1501, HLA-DRB1*0301) to predict CTL epitopes and HTL epitopes [29]. CTL epitopes were predicted using IEDB(http://tools. immuneepitope.org/) [30] and NetCTLpan-1.1(https://services.healthtech.dtu.dk/service.php? NetCTLpan-1.1) with an epitope length of 9, and overlapping sequences were screened [31]. It is important to note that NetCTLpan-1.1 indexing starts from 0, so adding 1 is necessary for alignment with IEDB sequences. For HTL epitope prediction, NetMHCIIpan-4.0 (https:// services.healthtech.dtu.dk/service.php?NetMHCIIpan-4.0) [32] and IEDB tools were employed. NetMHC-IIpan-4.0 was configured with the original threshold and an epitope length of 15, with dominant epitopes selected based on overlapping sequences.

## 2.5 Prediction of protein B-cell epitopes

B cell epitope includes linear epitope and conformational epitope.SVMtrip (sysbio.unl.edu/ SVMTriP/prediction.php) was used to predict linear B cell epitopes, and the epitope length was set to 20 [33]. The conformational epitope was predicted by Ellipro of IEDB.

## 2.6 T-cell epitopes molecular docking to HLA alleles

To examine the relationship between T cell epitopes and alleles, we used the HDOCK(http:// hdock.phys.hust.edu.cn/) server to select class HLA-I (HLA-A * 02:01) and class HLA-II (HLA-DRB1 * 07:01) alleles for molecular docking with T cell epitopes. Finally, the LIGPLOT (v.2.2.8) was used to evaluate the interactions between epitopes and various residues of MHC alleles.

## 2.7 mRNA vaccine structure design

To produce mRNA vaccines, highly antigenic, non-allergic, and non-toxic epitopes are connected using linkers such as AAY, GPGPG, and KK for CTL, HTL, and B-cell epitopes, respectively [20]. In order to facilitate the detection and purification of the recombinant protein later, we added a set of HHHHHH sequences to the C-terminal of the amino acid sequence and connected them with GGGS linkers [34]. The 5'-cap structure, essential for mRNA translation, initiates mRNA translation and boosts mRNA stability and efficiency [35]. Translation and degradation efficiency of mRNA can be regulated by 5´- and 3´-UTRs [36]. The Kozak sequence includes a start codon [37]. The tPA Signal (UniProt ID: P00750) aids in secreting the signal sequence of epitopes, enhancing vaccine immunogenicity [38]. It is important to note that we used only the precursor sequence of the tPA signal peptide, which can be transcribed but not translated. Adjuvant resuscitation promoting factor (RpfE) (Rv2450c) can improve adaptive immune response. CTL epitopes are guided to the MHC-I region of the endoplasmic reticulum by the 3' MITD (Uniprot ID: Q8WV92) [39]. The addition of a Poly (A) tail can facilitate protein translation and enhance mRNA stability [35]. These elements contribute to the successful construction of mRNA vaccines.

## 2.8 Prediction of homology between vaccine and humans

Non-homologous proteins can be reduced to stimulate the body to produce an adverse immune response [40]. By using NCBI BlastP (https://blast.ncbi.nlm.nih.gov/Blast.cgi) database, comparing vaccine and Homo sapiens (TaxID: 9606) to reduce the risk of autoimmune. An E value greater than 0.5 is considered non-homologous and suitable for vaccine construction [41].

## 2.9 Assessment of vaccine structure

An effective vaccine must possess appropriate physical and chemical properties. Parameters such as molecular weight, theoretical isoelectric point, amino acid composition, instability index, aliphatic index, and total average hydrophilicity (GRAVY) were evaluated using Prot-Param. The isoelectric point (pI) indicates the pH at which a molecule or surface is neutral and impacts solubility at specific pH levels. The aliphatic index is a measure of a protein's thermal stability, with proteins considered unstable if their instability index exceeds 40. A positive GRAVY value signifies hydrophobicity, while a negative value indicates hydrophilicity [42]. Antigenicity, which refers to an antigen's ability to selectively bind to antibodies or lymphocytes, was assessed using VaxiJen2.0 during vaccine design. Furthermore, allergic reactions were assessed using AllerTOP2.0 to ensure the vaccine does not induce sensitization [43]. ToxinPred was utilized to evaluate the vaccine's toxicity, confirming that the constructed vaccine does not produce any toxic effects [44].

## 2.10 Immune response simulation

C-Immsim was utilized to model the three injection intervals of 1, 84, and 168 to replicate the immunological response induced by the vaccine in the body [45]. The analysis focused on high frequency alleles—HLA-A*1101, HLA-A*0201, HLA-B*5101, HLA-B*3501, HLA-DRB1*0701, and HLA-DRB1*1501—in the Xinjiang population. The simulation parameters included a random seed of 12345, a simulation volume of 50, a simulation step of 1050, and a dose of 1000 units [46].

## 2.11 Optimization of mRNA codons and in silico cloning

The online codon optimization server JCat tool was utilized for codon optimization and analysis [47]. Escherichia coli was chosen as the expression host [48] and *BamHI* and *XHOI* restriction sites were deliberately excluded to obtain the desired DNA sequences. PVAX1 was employed as the vector for electronic cloning, with primers being added and amplified using Snap Gene. The GC content ranged between 30% and 70%, while the primer length was set between 15–30 bp. It was ensured that the upper and lower annealing temperatures were similar. Finally, the amplification process was completed using *BamHI* and *XHOI* restriction enzymes.

## 2.12 Agarose gel electrophoresis

Agarose gel electrophoresis of target genes (post-PCR), vectors, and recombinant plasmids in Snap Gene was done in a TBE buffer at 1% agarose concentration.

## 2.13 Prediction of secondary structure of mRNA vaccine

To ensure the effectiveness of mRNA transcription, we used the RNAfold(http://rna.tbi.univie.ac.at/cgi-bin/RNAWebSuite/RNAfold.cgi)server to predict the secondary structure of single-stranded RNA sequences. Currently, it can handle sequences up to 10,000 nt for minimum free energy predictions and up to 7,500 nt for partition function computations. The server also predicts the center of mass secondary structure and minimal free energy (MFE) of mRNA.

## 2.14 Prediction of secondary and tertiary structure of vaccines

The specific conformation formed by the polypeptide backbone atoms spiraling or folding along a defined axis is known as a protein's secondary structure. To ensure that the vaccine proteins are translated correctly, we analyze the distribution of α helix, β sheet, random coil,

and extended chain components of the vaccine, we utilized the SOPMA(http://npsa-pbil.ibcp.fr/cgibin/npsa_automat.pl?page=/NPSA/npsa_sopma.html) server. The SWISS-MODEL (https://swissmodel.expasy.org/assess) tool was employed to predict the stoichiometry and overall structure of the complex through homology modeling, involving steps such as input data submission, template search, template selection, modeling, and model quality assessment. Subsequently, the predicted tertiary structure was refined using the GalaxyWEB(http://galaxy.seoklab.org/) server, which enhances the core structure based on multiple templates and optimizes unreliable loops or termini. This refinement method is recognized for its high performance in template-based modeling [49]. The final 3D model was rendered using Discovery Studio(2019) software.

## 2.15 Quality testing of models

The PROCHECK(https://saves.mbi.ucla.edu/) server was utilized to assess the quality of the tertiary structure, which indicates the level of agreement between the model structure and the experimental data, along with the geometric features. The server mainly includes PROCHECK, WHATCHECK,ERRAT,Verify-3D and PROVE five commonly used tests. PROCHECK analysis takes high-resolution crystal structure parameters in PDB as reference, and gives a series of stereochemical parameters of the submitted model. After the analysis was completed, the percentage of amino acids in the "most favoured regions"," additional allowed regions", "generously allowed regions ", and" disallowed regions" were listed in the Ramachandran Plot line. It is generally required that the amino acid residues in the optimal region account for more than 90% of the whole protein, and the number of amino acids in the disallowed region should be less than 5% of the total amino acids. It is considered that the conformation of the model conforms to the rules of stereochemistry [50]. Additionally, ProSA (https://prosa.services.came.sbg.ac.at/prosa.php) was employed to validate the protein structure, ProSA is a tool for examining potential errors in 3D structural models of proteins, which uses X-ray analysis, NMR spectroscopy, and theoretical calculations for structural verification of proteins. From this we can get the z-score and its residue energy graph. The z-score represents the overall mass of the model and measures the energy distribution deviation of the total energy of the structure relative to the random conformation. A z-score outside the characteristic range of a natural protein indicates a faulty structure. Energy diagram show local model mass by plotting energy as a function of amino acid sequence position. In general, a positive value corresponds to a problem or error in the model [51].

## 2.16 Molecular docking

MEV activates the immune system as an antigen, triggering an immune response. In our study, we utilized the HDOCK server to perform molecular docking of MEV with TLR4 (PDB ID: 3FXI). HDOCK employs intrinsic scoring functions for protein-protein and protein-DNA/RNA docking, streamlining the docking process. The procedure involves inputting the FASTA format of MEV and TLR-4, conducting a sequence similarity search to identify homologous sequences in the PDB database, generating homologous templates for the receptor and ligand, comparing these templates to select the best ones, and performing modeling and comparison using Modler and ClustalW, respectively. Global docking is then carried out using the FFT docking program to predict the binding orientation, and the most favorable model is selected from the generated models [52].

## 2.17 Molecular dynamics simulation

In this study, Gromacs2022.3 was used to simulate the molecular dynamics (MD) of MEV-TLR4 complex [53]. The water molecule (Tip3p water model) served as the solvent in the simulation, which was run at a static temperature of 300K and atmospheric pressure of 1 Bar. The overall charge of the simulated system was neutralized by adding the proper number of Na+ ions. The force field used in the simulation was Amber99sb-ildn. Using a coupling constant of 0.1 ps and a duration of 100 ps, the molecular dynamics simulation system first uses the steepest descent method to minimize the energy. It then performs the isothermal isobaric ensemble (NPT) equilibrium and isothermal isocorph system (NVT) equilibrium for 100,000 steps, respectively. Lastly, a simulation using free molecular dynamics was run. The entire operation took 100ns and involved 5000000 steps, each with a step length of 2fs. Following the completion of the simulation, the trajectory was analyzed using the software's built-in tool to calculate metrics such as root-mean-square deviation (RMSD), root-mean-square fluctuation (RMSF), hydrogen bonds, protein Radius of Gyration(Gyrate) and Solvent Accessible Surface Area(SASA) for each amino acid trajectory. These calculations were then combined with data on free energy (MMGBSA), free energy topography, and other relevant information.

# 3. Results

## 3.1 Selection of target proteins

The UniProt search numbers for *PknD* and *Rv0986* proteins are P9WI79 and P9WQK1, respectively. The amino acids of *PknD* and *Rv0986* are 664aa and 248aa, respectively. In silico methods were used to analyze the antigenicity and sensitization of these proteins, resulting in antigenicity values of 0.5633 for *PknD* and 0.4736 for *Rv0986*, both exceeding 0.4. It was determined that both *PknD* and *Rv0986* are non-allergens. Furthermore, the physicochemical properties (Table 1) of *PknD* and *Rv0986* were examined, indicating that they are hydrophilic stable proteins.

## 3.2 Sequence retrieval

The high accuracy (Fig 2) of MAFFT in jalview indicates that the two proteins share several homologies, suggesting that they may derive from the same gene and have similar roles in the immune response.

**Table 1. Physicochemical properties of amino acids.**

| Amino acids | Serial number | Antigenicity | Allergenicity | Instability index (II) | Grand average of hydropathicity (GRAVY) |
|---|---|---|---|---|---|
| PknD | P9WI79 | 0.563 | non-allergen | 26.84 | -0.055 |
| Rv0986 | P9WQK1 | 0.473 | non-allergen | 32.44 | -0.625 |

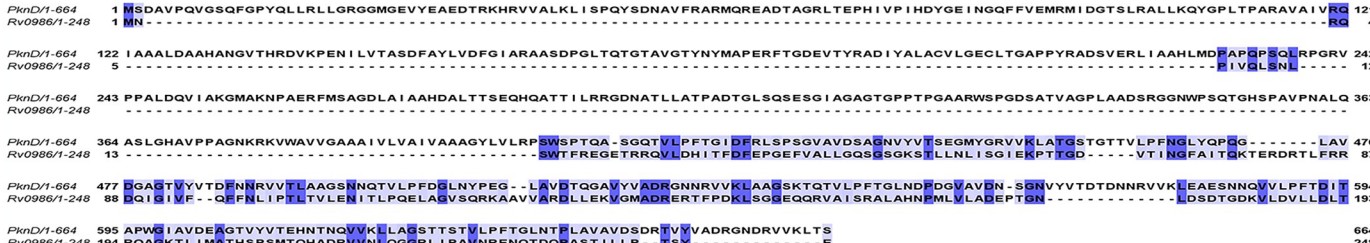

**Fig 2. Homologous sequence alignment of proteins.** The blue portions represent similar amino acid sequence (The darker the blue, the more conservative it is).

### 3.3 Prediction of signal peptides

While we used SignalP6.0 and Signal BLAST to predicte signal peptide of the *PknD* and *Rv0986*, there was no signal peptide in the end. Therefore, *PknD* and *Rv0986* proteins are not directed to other suborganelles of the cell or secreted to play roles outside the cell after synthesis (Fig 3) (S1 and S2 Figs).

### 3.4 Prediction of T-cell epitopes

Select two of the top 10 epitopes of the software and pick the overlapping sequence. The antigenicity of epitopes was analyzed by VaxiJen. Allergen v1.0 was used to predict epitope sensitization; Use ToxinPred to predict epitope toxicity. In the end, we obtained 3 CTL dominant epitopes and 11 HTL dominant epitopes (Tables 2 and 3). These epitopes are highly antigenic, non-allergenic and non-toxic (S1–S8 Tables).

### 3.5 Prediction of B-cell epitopes

Three linear epitopes of 20 amino acids were obtained by analysis (S9 Table). For conformational epitopes, we finally selected two discontinuous epitopes with a length greater than 5 amino acids. The predicted score of *Rv0986* is 0.812, and the predicted score of *PknD* is 0.681 (Tables 4 and 5) These epitopes are highly antigenic, non-allergenic and non-toxic (Fig 4).

### 3.6 Molecular docking of T-cell epitopes to HLA alleles

We performed molecular docking simulations to demonstrate the interaction of HLA alleles with selected T cell epitopes. Results for ITAPWGIAV(CTLs) interacting with HLA-A*02:01, Docking Score:-171.21 Confidence Score:0.6045 ligand RMSD (Å):109.11. FQFFNLIPTLTVLEN(HTLs) interacting with HLA-DRB1*07:01 Results, Docking Score-123.45 Confidence Score:0.3703 ligand RMSD (Å):55.09. These results indicate a high affinity for the docking complex (Figs 5 and 6).

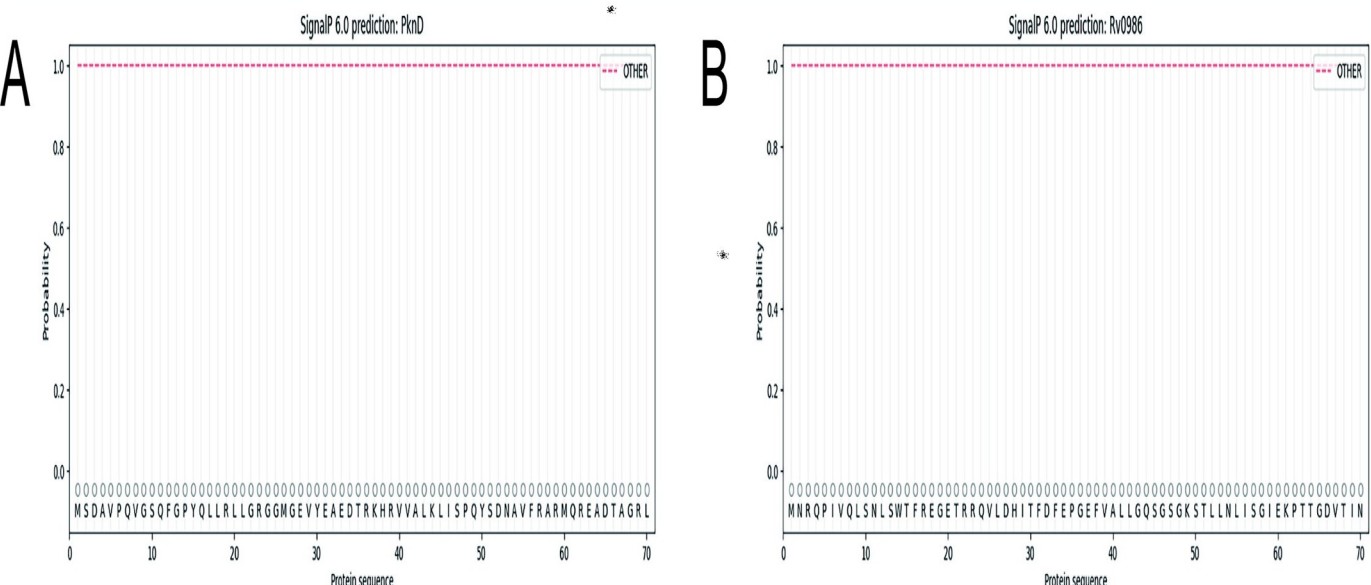

**Fig 3. (A-B) Protein signal peptide was analyzed by SignalP6.0.** Other: the probability that the sequence does not have any signal peptides.

**Table 2. Physical properties, antigenicity and scores of CTL dominant epitopes.**

|  | Serials | CTL epitopes | Molecular weight | Instability index | Grand average of hydropathicity (GRAVY) | Theoretical PI | Antigenicity | Score |
|---|---|---|---|---|---|---|---|---|
| Rv0986 | 114–122 | ELAGVSQRK | 987.12 | 30.29 | -0.756 | 8.85 | 1.0888 | 0.492 |
|  | 14–22 | WTFREGETR | 1181.27 | -2.48 | -1.767 | 6.14 | 1.515 | 0.423 |
| PknD | 593–601 | ITAPWGIAV | 927.11 | 6.13 | 1.467 | 5.52 | 0.939 | 0.731 |

**Table 3. Physical properties, antigenicity and scores of HTL dominant epitopes.**

|  | Serials | HTL epitopes | Molecular weight | Instability index | Grand average of hydropathicity (GRAVY) | Theoretical PI | Antigenicity | Score |
|---|---|---|---|---|---|---|---|---|
| Rv0986 | 199–213 | TLIMATHSPSMTQHA | 1625.88 | 52.75 | 0.033 | 6.61 | 0.586 | 0.388 |
|  | 94–108 | FQFFNLIPTLTVLEN | 1796.10 | 2.39 | 0.767 | 4.00 | 0.650 | 0.409 |
|  | 93–107 | VFQFFNLIPTLTVLE | 1781.12 | 2.39 | 1.280 | 4.00 | 0.665 | 0.317 |
|  | 150–164 | GEQQRVAISRALAHN | 1649.83 | 52.95 | -0.633 | 9.61 | 0.442 | 0.295 |
|  | 70–84 | NGFAITQKTERDRTL | 1749.94 | 2.10 | -1.100 | 8.75 | 0.493 | 0.323 |
|  | 174–188 | TGNLDSDTGDKVLDV | 1548.62 | -16.15 | -0.56 | 3.77 | 0.890 | 0.378 |
| PknD | 421–435 | GIDFRLSPSGVAVDS | 1519.67 | 43.60 | 0.333 | 4.21 | 2.029 | 0.774 |
|  | 123–137 | AAALDAAHANGVTHR | 1474.60 | -12.01 | -0.013 | 6.96 | 1.208 | 0.685 |
|  | 420–434 | TGIDFRLSPSGVAVD | 1533.70 | 25.10 | 0.340 | 4.21 | 2.000 | 0.753 |
|  | 590–610 | PWGIAVDEAGTVYVT | 1577.75 | -31.53 | 0.513 | 3.67 | 0.716 | 0.871 |
|  | 510–524 | NYPEGLAVDTQGAVY | 1596.71 | -6.61 | -0.260 | 3.67 | 0.487 | 0.755 |

**Table 4. Physical properties, antigenicity and scores of LBEs dominant epitopes.**

|  | Serials | LBEs epitopes | Molecular weight | Instability index | Grand average of hydropathicity (GRAVY) | Theoretical PI | Antigenicity | Score |
|---|---|---|---|---|---|---|---|---|
| Rv0986 | 61–80 | KPTTGDVTINGFAITQKTER | 2177.44 | -4.42 | -0.72 | 8.59 | 0.942 | 0.562 |
| PknD | 50–69 | YSDNAVFRARMQREADTAGR | 2314.52 | -6.03 | -1.13 | 8.74 | 1.043 | 1.000 |
|  | 87–106 | QFFVEMRMIDGTSLRALLKQ | 2383.85 | 14.94 | 0.125 | 8.75 | 0.5899 | 0.914 |

**Table 5. Physical properties, antigenicity and scores of CBEs dominant epitopes.**

|  | Serials | Residues | CCBEs | Molecular weight | Instability index | Grand average of hydropathicity (GRAVY) | Theoretical PI | Antigenicity | Score |
|---|---|---|---|---|---|---|---|---|---|
| Rv0986 | 228–248 | A:V228,A:N229,A:R230,A:E231,A:N232,A:Q233,A:T234,A:D235,A:Q236,A:P237,A:A238,A:S239,A:T240,A:I241,A:L242,A:L243,A:P244,A:T245,A:S246, A:Y247,A:E248 | VNRENQTDQPASTILLPTSYE | 2376.56 | 39.03 | -0.910 | 4.14 | 0.3140 | 0.812 |
| PknD | 242–267 | A:D242,A:S243,A:D244,A:R245,A:T246,A:T261,A:S262,A:L263,A:E264,A:H265,A:H266,A:H267 | DS DSDRTTSLEHHH | 1434.44 | 12.65 | -1.983 | 5.73 | 0.6945 | 0.681 |

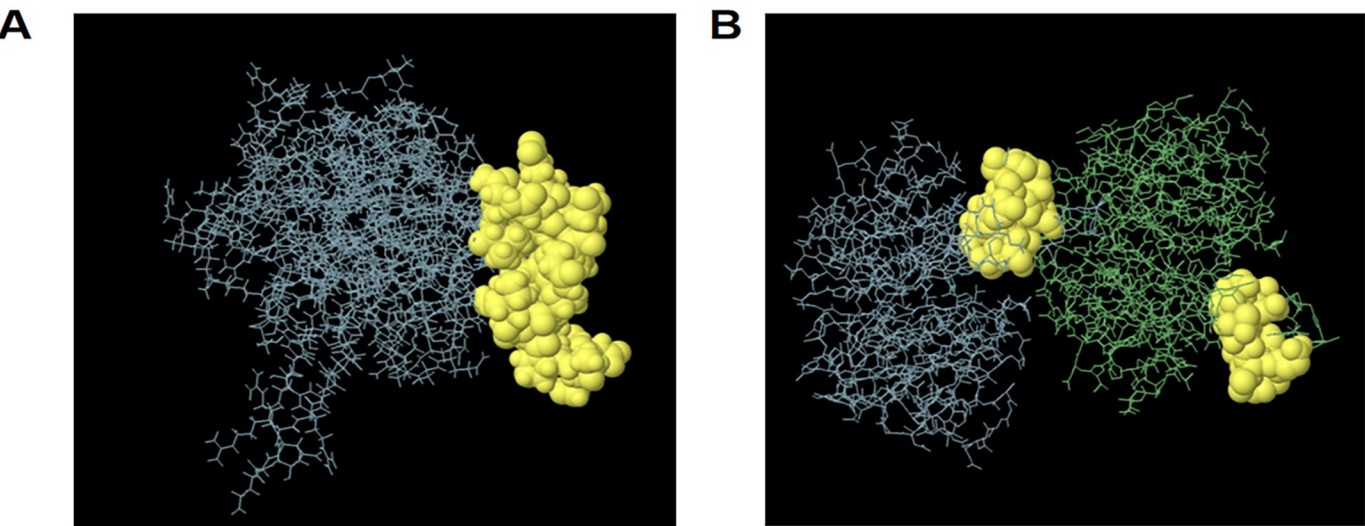

**Fig 4. B cell conformational epitope predicted by Ellipro.** The yellow spheres represent B cell conformational epitopes, the gray area represents the majority of the polyprotein. (A)Rv0986 conformational epitope residues:**V228,N229,R230, E231,N232,Q233,T234,D235,Q236,P237,A238,S239,T240,I241,L242,L243,P244, T245,S246,Y247,E248.** (B)PknD conformational epitope residues:D242,S243,D244,R245,T246,T261,S262,L263, E264,H265,H266,H267.

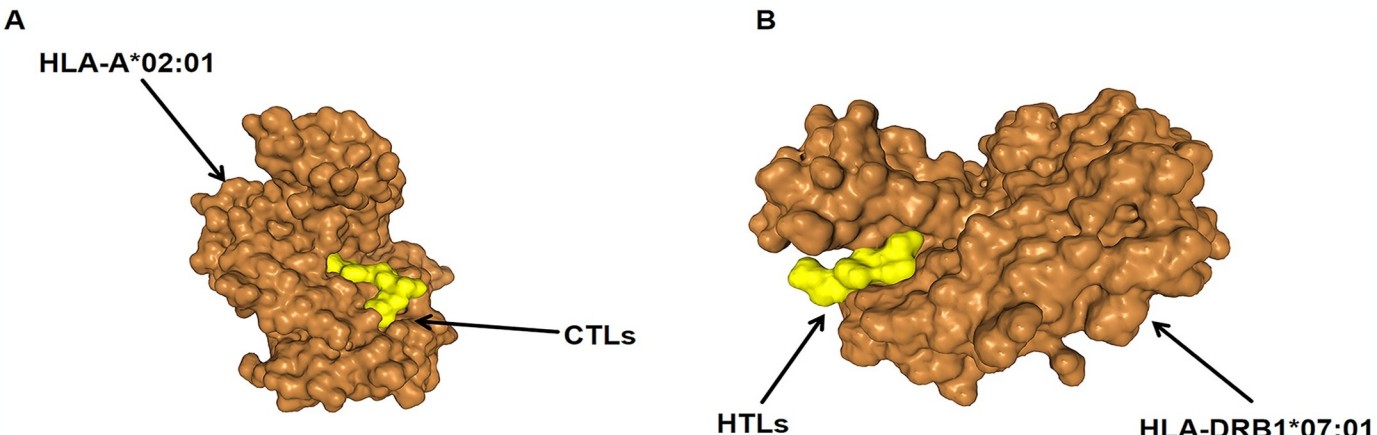

**Fig 5. Docking complex display (A)Molecular docking between ITAPWGIAV(CTLs) and HLA-A*02:01(B)Molecular docking between FQFFNLIPTLTVLEN(HTLs) and HLA-DRB1*07:01.**

## 3.7 Construction of novel mRNA vaccines

The mRNA vaccine is constructed from the N to C terminus as follows:5'm7GCap-5'UTR-Kozak sequence-tPA(Signal peptide)-EAAAK Linker-RpfE(Adjuvant)-`AAYELAGVSQRKAAYW TFREGETRAAYITAPWGIAVGPGPGTLIMATHSPSMTQHAGPGPGFQFFNLIPTLTVLE NGPGPGVFQFFNLIPTLTVLEGPGPGGEQQRVAISRALAHNGPGPGNGFAITQKTERDR TLGPGPGTGNLDSDTGDKVLDVGPGPGGIDFRLSPSGVAVDSGPGPGAAALDAAHANGVT HRGPGPGTGIDFRLSPSGVAVDGPGPGPWGIAVDEAGTVYVTGPGPGNYPEGLAVDTQ GAVYKKKPTTGDVTINGFAITQKTERKKYSDNAVFRARMQREADTAGRKKQFFVEMRMID GTSLRALLKQKKVNRENQTDQPASTILLPTSYEKKDSDRTVYVADRGNDRVVKLTSLEHHH GGGSHHHHHH−MITD` sequence-Stop codon-3'UTR-Poly(A)tail. The selected epitopes are

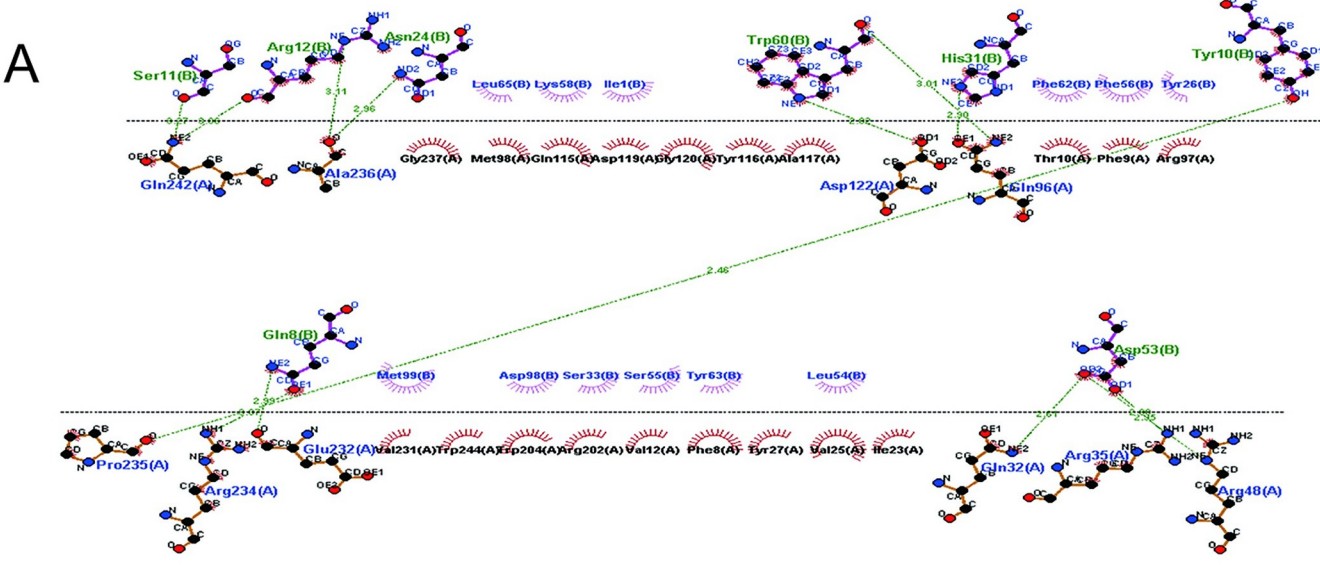

**Fig 6. (A-B)Epitopes and their corresponding MHC allele interaction using the LIGPLOT.** Hydrogen bonds are represented by dotted green lines, and red semicircular circles indicate residues involved in hydrophobic interactions.

connected using three linkers, AAY,GPGPG and KK, which function independently. The linker AAY is breakable, GPGPG is rigid, and KK is flexible (Fig 7). Finally, after homology comparison, we found that no significant similarity was found between the vaccine and the host, and the E value was less than 0.05, indicating that the vaccine we constructed was reasonable (S3 Fig).

### 3.8 Evaluate the physicochemical properties, antigenicity, allergenicity and toxicity of the MEV

The MEV protein consists of 383 amino acids, with a molecular weight of 40.196 kDa and a theoretical isoelectric point of 8.65. It contains 36 acidic amino acids (Asp+Glu) and 38 basic

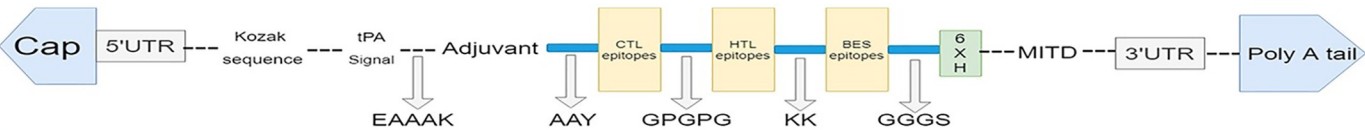

**Fig 7. Vaccine construct from N-terminal to C-terminal.**

amino acids (Arg+Lys). The formula for MEV is $C_{1773}H_{2765}N_{521}O_{541}S_5$, with a total of 5,605 atoms. The instability index (II) is 7.49, indicating stability. The aliphatic index is 66.50 and the Grand average of hydropathicity (GRAVY) is -0.472, suggesting hydrophilicity. Furthermore, the vaccine exhibits antigenicity, non-Allergen, and non-Toxicity. In conclusion, the vaccine design appears to be feasible. Finally, (Table 6) presents all the final results.

### 3.9 Immune response simulation

Utilizing C-ImmSim online software, we simulated the immune response to three vaccine injections. The results demonstrated a notable increase in the immune response to the second and third injections as compared to the first. As the antigen content decreased, there was a gradual rise in IgM and IgG levels, with IgM consistently surpassing IgG (Fig 8A). This trend can be attributed to the development of immune memory post-antigen exposure, leading to an increase in CTLS and HTL numbers (Fig 8B–8D). Similarly, the presence of memory cells resulted in an augmentation of B cells, crucial for humoral immunity (Fig 8E and 8F). Dendritic cell numbers remained constant (Fig 8G), while macrophage numbers showed a gradual increase (Fig 8H). Moreover, levels of IFN-γ, TGF-β, and IL-2 increased, with a lower Simpson index (D) indicating immune response variations (Fig 8I). These results show that the dose and time interval of injection are reasonable.

### 3.10 mRNA codon optimization and in silico cloning

Codon optimization tools play a crucial role in enhancing the translation of mRNA vaccines within host cells. The quality of codon optimization is typically evaluated based on the codon adaptation index (CAI) and GC content. A CAI value of 1.0 signifies optimal optimization,

**Table 6. The physiochemical profiling of the mRNA vaccine.**

| Physiochemical profiling | Measurement | Indication |
|---|---|---|
| Number of amino acids | 383 | Appropriate |
| Molecular weight | 40195.92 | Appropriate |
| Theoretical pI | 8.65 | Basic |
| Total number of negatively charged residues (Asp + Glu) | 36 | - |
| Total number of positively charged residues (Arg + Lys) | 38 | - |
| Formula | $C_{1773}H_{2765}N_{521}O_{541}S_5$ | - |
| Total number of atoms | 5605 | - |
| Instability index (II) | 7.49 | Stable |
| Aliphatic index | 66.50 | Thermostable |
| Grand average of hydropathicity (GRAVY) | -0.472 | Hydrophilic |
| Allergenicity | NON-ALLERGEN | Non-Allergen |
| Antigenicity | 0.9510 | Antigenic |
| Toxicity | Non-Toxic | Non-Toxic |

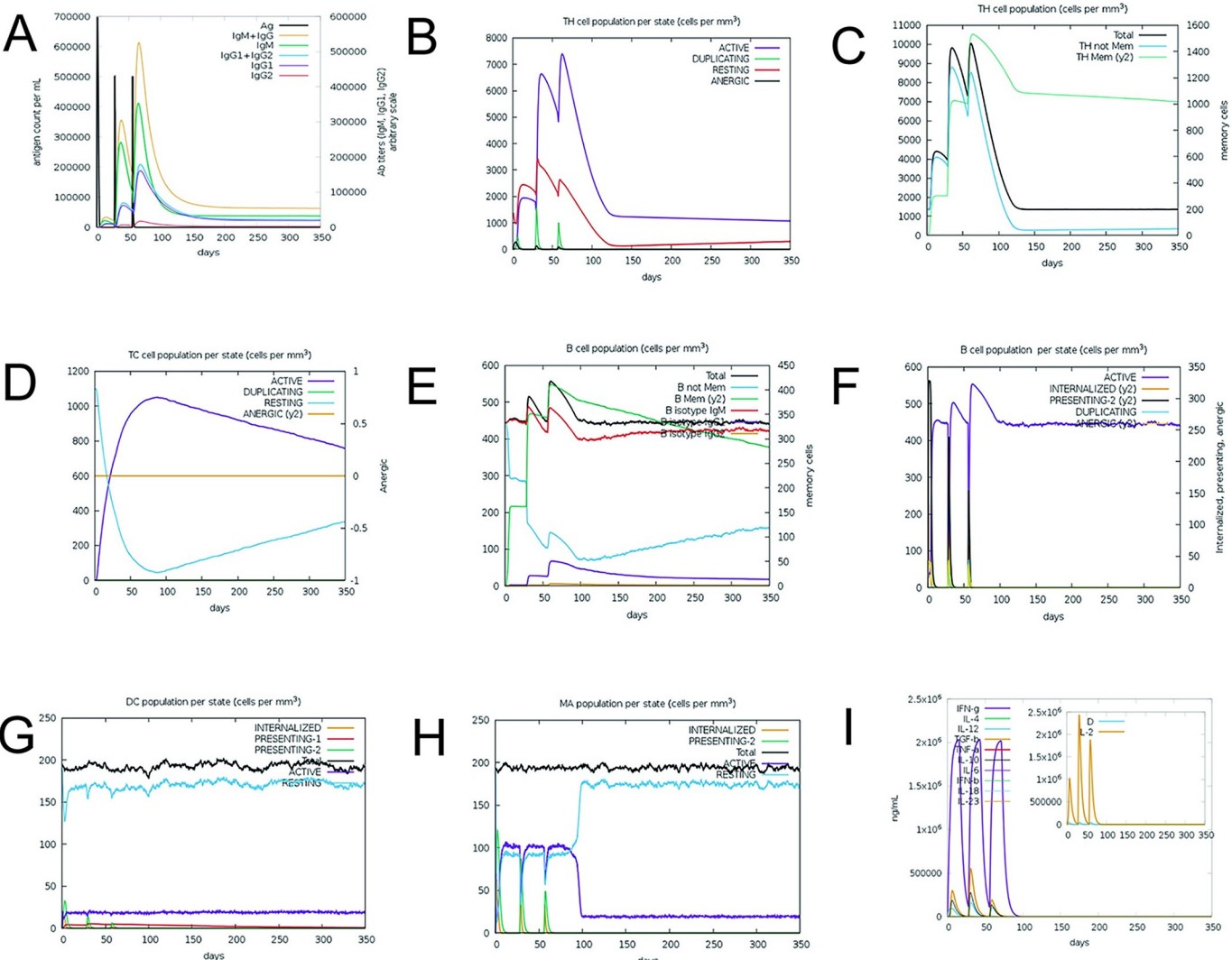

**Fig 8.** (A)Immunoglobulins in various states (B)The Helper T Cell Population in various states (C)The Helper T Cell Population in various states (D)The Cytotoxic T Cell Population in various states (E)The B cell population in various states (F)The B Cell Population in various states (G)Dendritic Cell Population in various states (H)Macrophage Population in various states (I)Cytokines and Interleukins Production with Simpson Index of the immune response.

with values above 0.8 considered favorable. Moreover, a GC content between 30–70% is known to facilitate effective gene expression in human hosts. Post-optimization, the average GC content was determined to be 56.74%. Following primer design principles, we developed an upstream primer (5'-GGATCCGCTGCTTACGAACTGGCTGGT-3') with a length of 27, an aTm value of 68, and a GC content of 59%, incorporating the *BamHI* enzyme restriction site at the 5' end. Similarly, the downstream primer (5'-CTCGAGGTGGTGGTGGTGGTGGTGAGAA-3') was designed with a length of 28, an aTm value of 66, and a GC content of 61%, featuring the *XhoI* enzyme restriction site at the 3' end. Subsequently, the target gene was amplified using Snap Gene, and for cloning purposes, the eukaryotic expression vector PVAX1 was selected. The amplified target gene successfully inserted into the multiple cloning site (MCS) region of the plasmid after removing the previously designed primer restriction site (Fig 9).

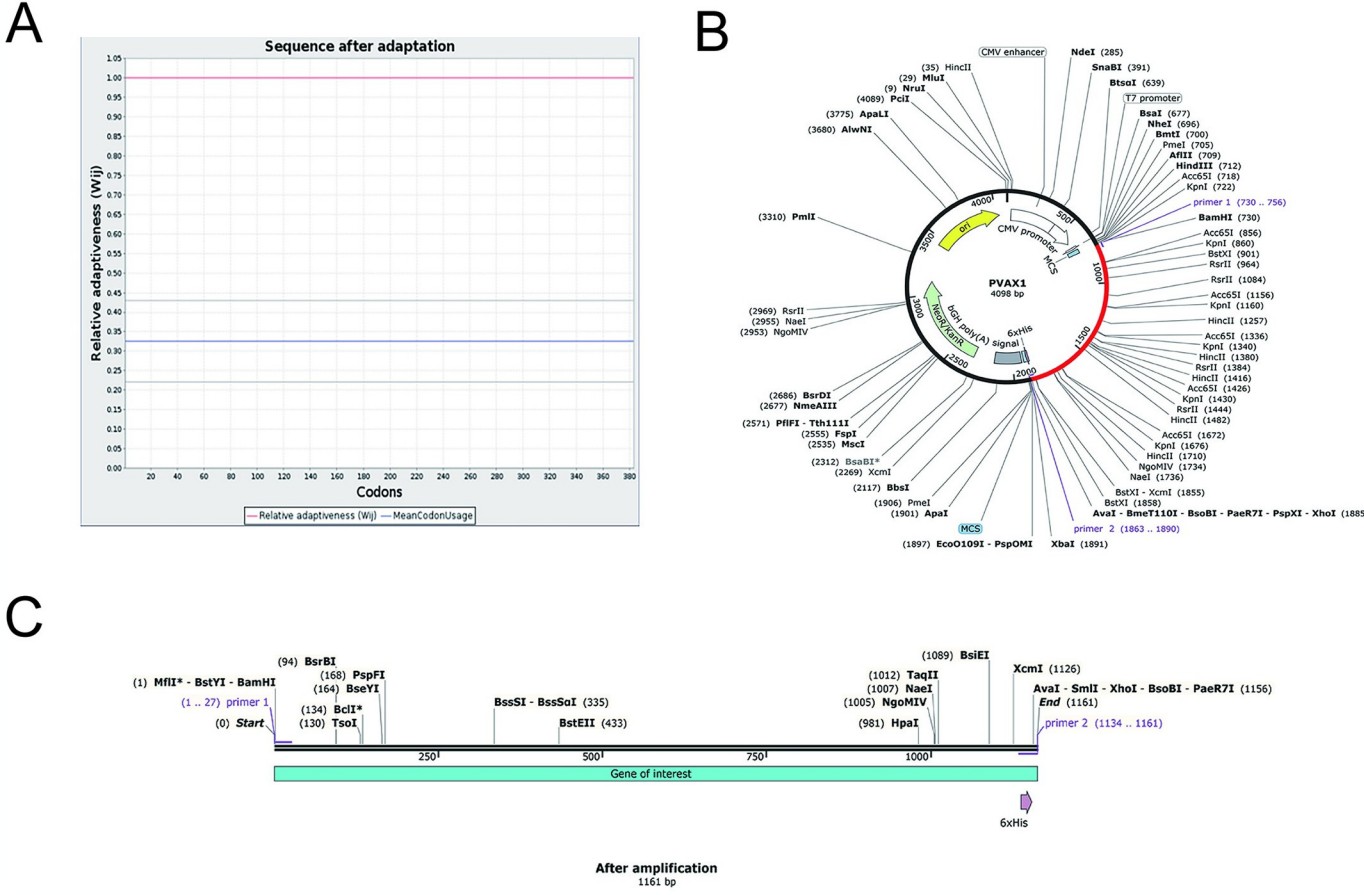

**Fig 9. Codon optimization and plasmid vector construction (A)Sequence after adaptation (B)The cloned MEV was inserted into the PVAX1 vector (C)After amplified(1161bp).**

### 3.11 Agarose gel electrophoresis

The amount of DNA was consistent with what had been predicted. The amplified sequence of MEV was 1161bp, the vector size of PVAX1 was 2999bp and the recombinant plasmid was 4098bp (Fig 10).

### 3.12 Prediction of secondary structure of mRNA vaccine

The input was optimized and submitted to the RNAfold server to predict the mRNA structure and free energy. It was observed that the mRNA exhibited the most favorable secondary structure with a minimum free energy (MFE) of -428.80 kcal/mol, while the centroid secondary structure had a minimum free energy of -288.01 kcal/mol (Fig 11).

### 3.13 Prediction of secondary and tertiary structure of vaccines

The predicted secondary structure of the vaccine revealed that 8.36% consisted of α-helix, 8.62% of β-angle, 48.83% of irregular curling, and 34.20% of extended chain, aligning well with the tertiary structure. The tertiary structure model of the vaccine was constructed using SWISS-MODEL software, further refined on GalaxyWEB, and visualized with Discover

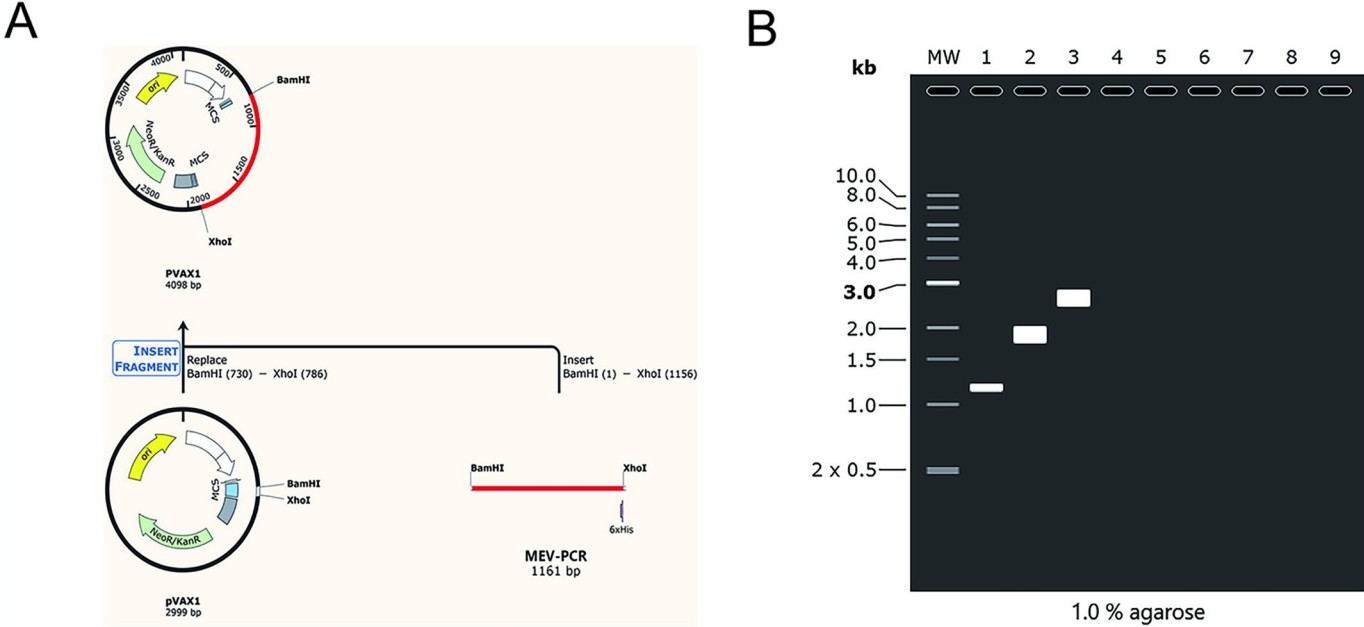

**Fig 10.** (A)In silico cloning simulation. Codon-optimized multiepitope sequences (red) inserted between the restriction sites *BamHI* and *XhoI* in the PVAX1 expression vector (black). (B)Simulated agarose gel electrophoresis results. "1" stands for MEV-PCR,"2"stand for PVAX1,"3" stand for recombinant plasmid.

Studio. In the visualization, gray indicates random curling, cyan indicates beta-folding, green indicates beta-turning, and red indicates alpha-spiraling (Fig 12).

### 3.14 Quality testing of models

We used PROCHECK to verify the validity of the tertiary structure, a server that can analyze its geometry and overall structure, and the Ramachandran diagram (Fig 13A) showed that

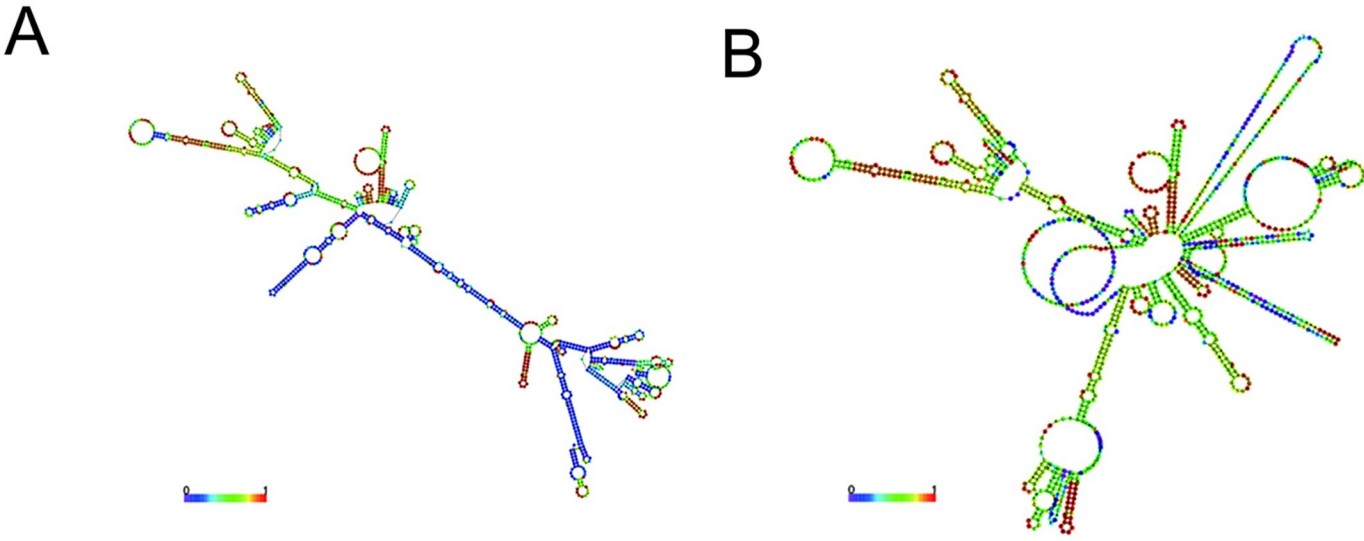

**Fig 11. (A)Optimal secondary structure (B)Centroid secondary structure.**

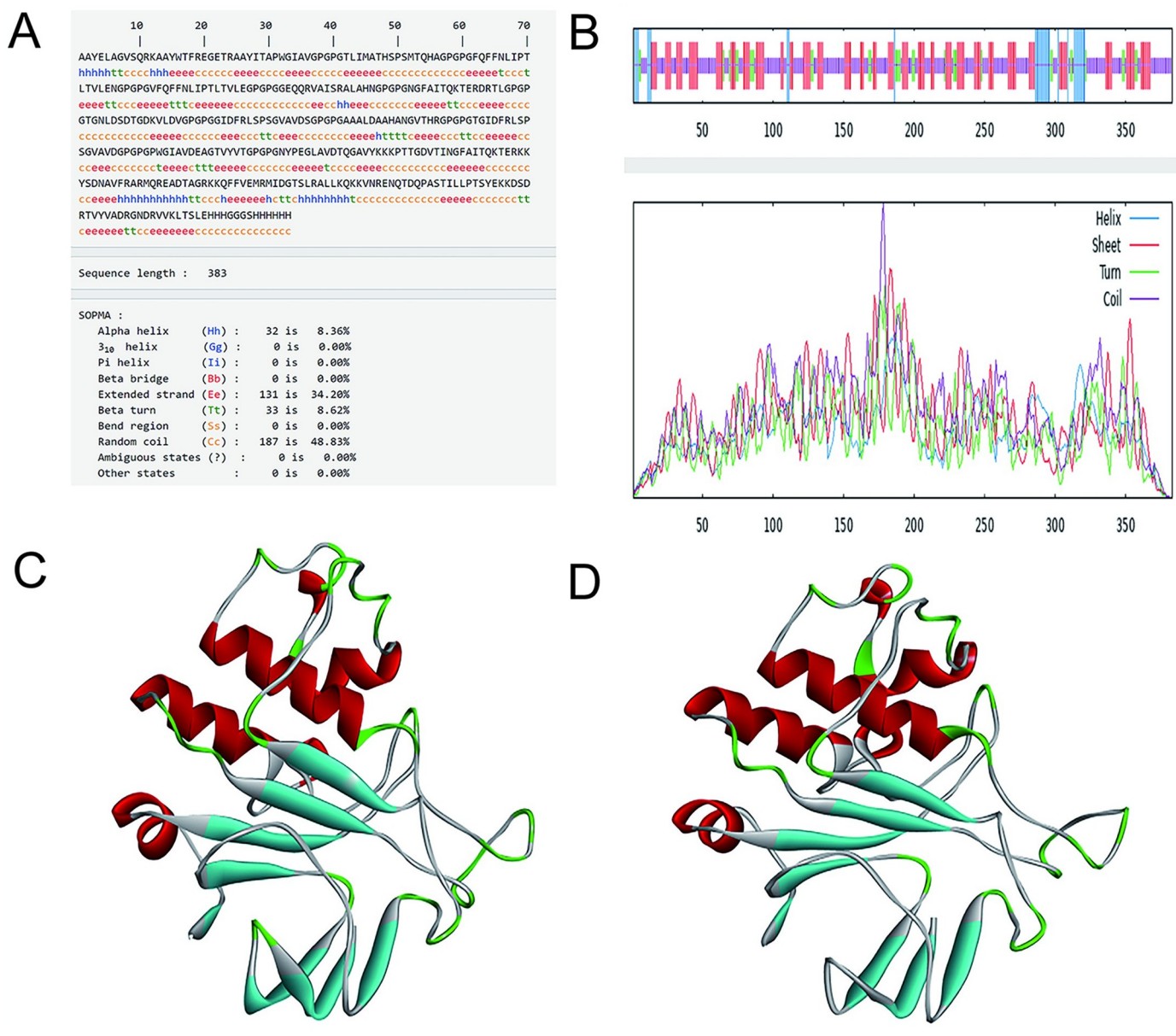

**Fig 12. (A-B)Prediction of vaccine secondary structure(C)Optimize the pre-tertiary structure(D)Optimize the post-tertiary structure.**

92.1% of the residues were in the most favoured regions and 6.5% were in the additional allowed regions and 0% were in the generously allowed regions and 1.4% were in the disallowed regions. The results were consistent with stereochemical rules, indicating that the vaccine structure was reasonable. Using ProSA-web, the Z-score (Fig 13B) was predicted to be -2.62, the energy diagram (Fig 13C) shows that most of the sequence positions are negative so the 3D structure we built was appropriate.

### 3.15 Molecular docking

Molecular docking is an essential technique in structure-based molecular design and screening as it forecasts the binding patterns and affinities between ligand and receptor molecules

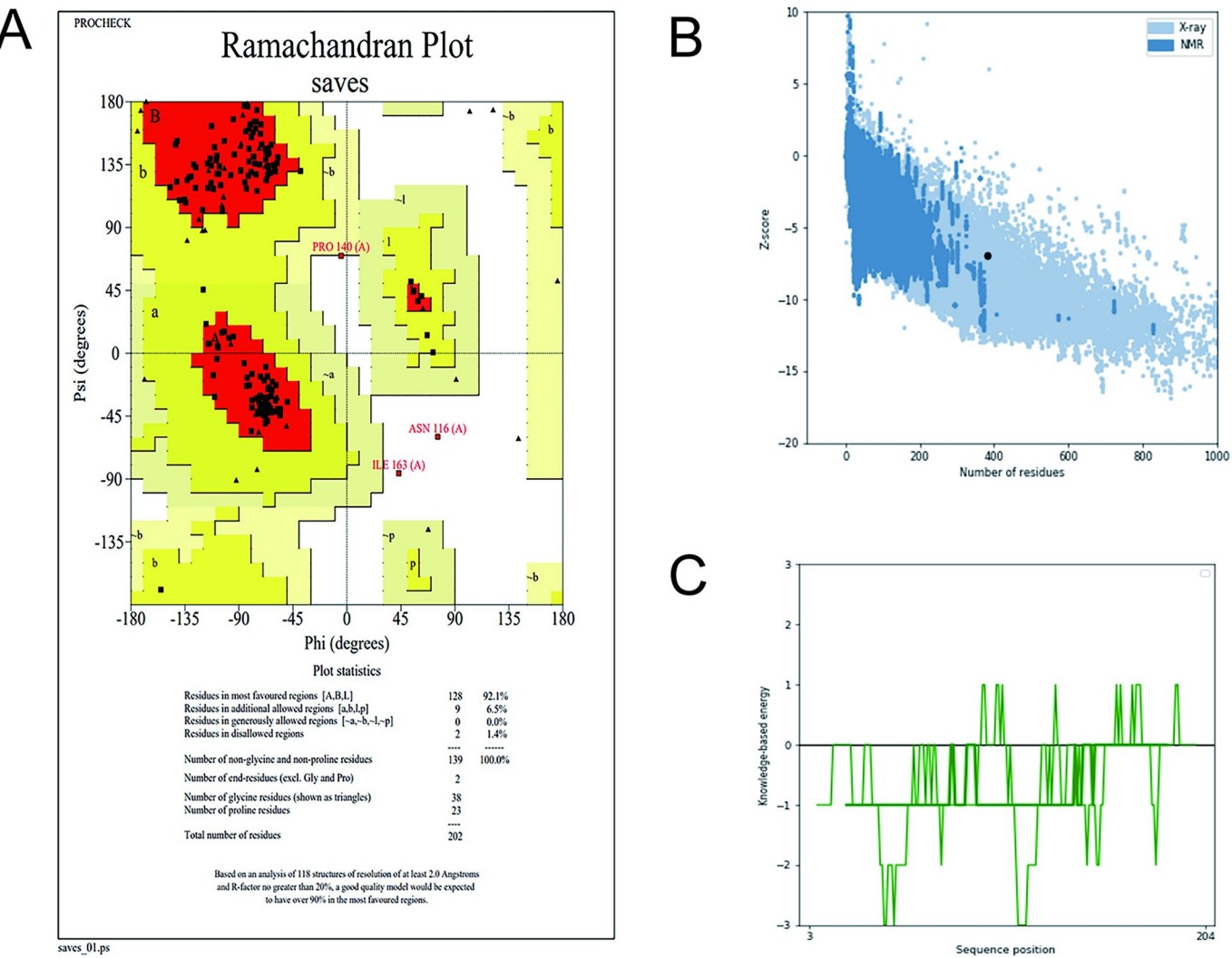

**Fig 13.** (A)The Ramachsndran diagram was analyzed using PROCHECK (B)Analyze the Z-score using the Pro-SA server. (C) Energy diagram using the Pro-SA server.

through interaction analysis. The HDOCK server was utilized for docking, with model 1 chosen for further analysis. The findings indicated a docking score of -321.20, ligand RMSD of 38.90A, and a confidence score of 0.9684 (Fig 14). The resulting structure was viewed using Discover Studio, and the intermolecular forces were visualized using PyMOL.

## 3.16 Molecular dynamics simulation

In this experiment, MEV-TLR4 interaction was simulated by Gromacs software (Fig 15), and the simulation results were analyzed. RMSD represents the distance between different structures and the same atom. A lower RMSD value indicates that the protein has high stability, while a higher RMSD indicates that the skeleton has undergone a conformational change during the simulation time.(Generally a range of less than 1 is normal) The blue Complex line in (Fig 15A) represents the RMSD after MEV-TLR4 docking. It can be seen that the docking

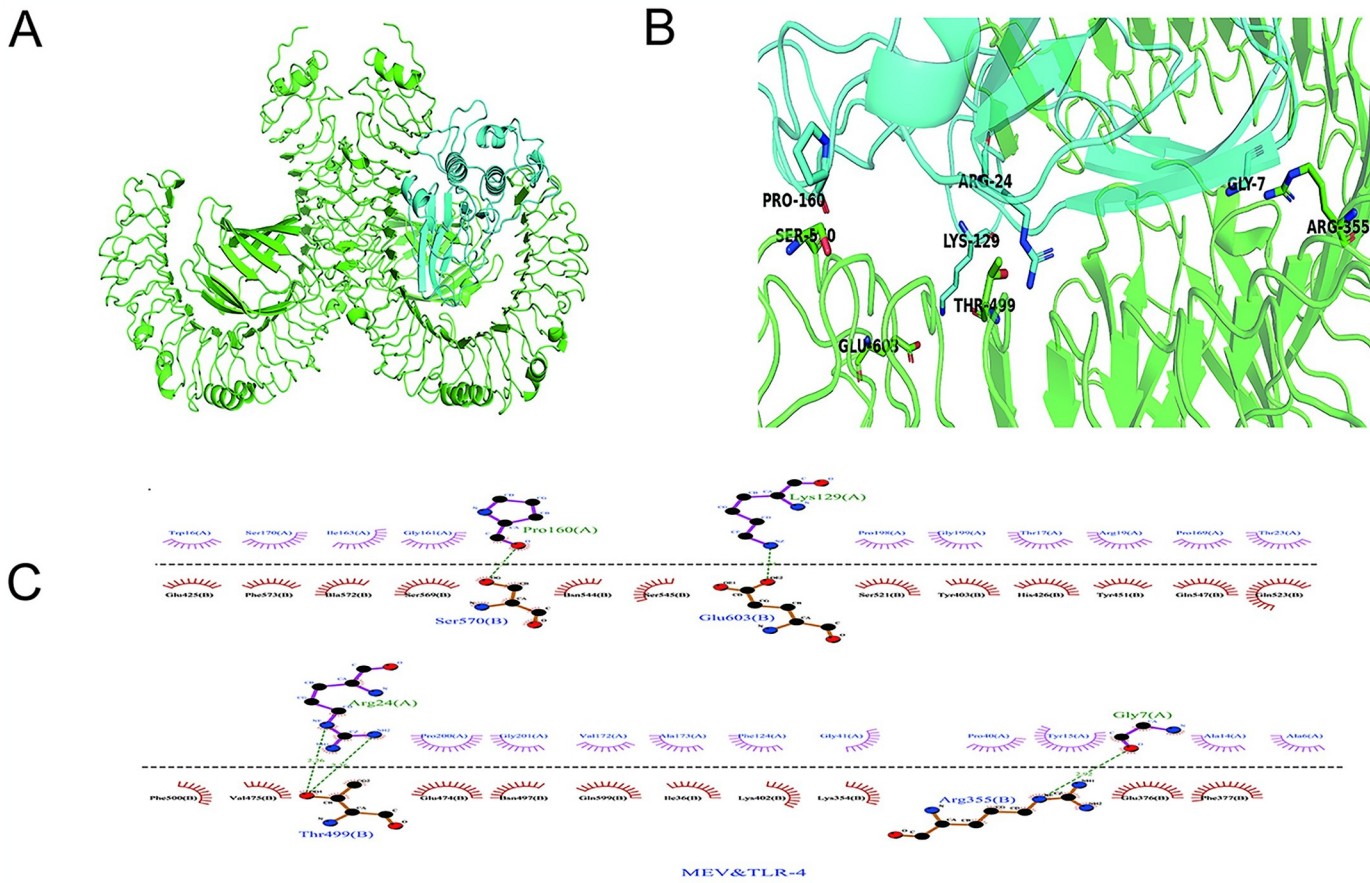

**Fig 14.** Molecular docking result (A)Docking complex, blue is the vaccine structure, green is the TLR4 receptor. (B)The interaction of MEV-TLR4 docking complex was demonstrated using PyMol. (C)The interaction of MEV-TLR4 complex and its 2D image were analyzed by Ligplot.

complex has been fluctuating in the range of 0.2–0.6, with the change range less than 1, and the overall change is stable. The results showed that the MEV-TLR4 interaction showed good characteristics and stability on the overall structure stability. RMSF indicates how flexible and vigorous the protein is throughout the simulation. This parameter determines the applicability of the ligand-receptor interaction over simulated time.(Fig 15B–15E)represents the receptor chain with a variation range of less than 0.3nm from the beginning to the end of the simulation, and (Fig 15F) represents the ligand chain with a variation range of 0.18nm from the beginning of the simulation to 0.32nm. The fluctuation is less than 0.3nm, which indicates that the protein-protein interaction has little effect on the stability of the internal structure of the protein molecule. Hydrogen bonding plays an important role in protein conformation preservation. The number of hydrogen bonds between (Fig 15G) remained between 2 and 14 during the simulation. This indicates that MEV-TLR4 interaction has good characteristics and stability. Gyrate was used to evaluate protein folding state. The Gyrate value between MEV-TLR4 in (Fig 15H) was about 4.12nm from the beginning to the end of the simulation. This suggests that the MEV-TLR4 interaction has little effect on the compactness of the overall structure of the protein molecule. SASA is used to assess the surface area of the protein molecules exposed to the solution and to assess the interactions between them and the stability of the protein structure. The SASA value in (Fig 15I) increased from the initial 715nm$^2$ to 725nm$^2$ during the

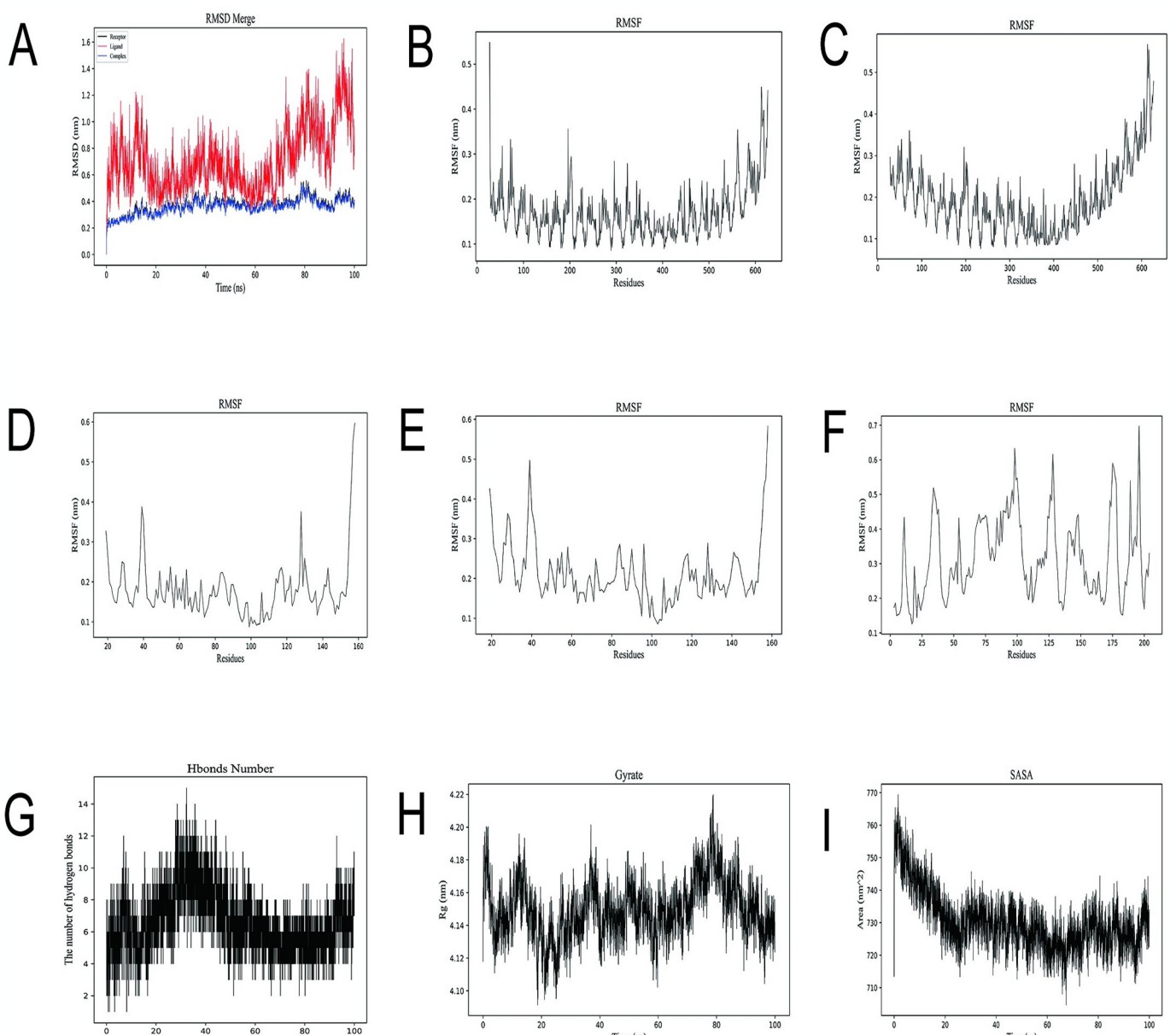

**Fig 15. Molecular dynamics simulation results.** (A) The RMSD locus of the receptor, ligand, and complex. The abscissa is the running time of MD simulation, and the ordinate is RMSD-value. (B-F) The RMSF locus of acceptor-ligand, the horizontal coordinate is the amino acid residue base in the docking complex, the ordinate is the rmsf value. (G-I) The trajectory of complex hydrogen bond, Gyrate and SASA respectively.

simulation, and the overall trend remained unchanged, indicating that MEV-TLR4 interaction had little effect on the surface characteristics and stability of protein molecules.

## 4. Discussion

Infants and young children are particularly susceptible to CNS TB, a debilitating and poorly understood illness. The antibiotic treatment regimen for CNS TB is based on the experience gained from treating TB. However, the increasing resistance to isoniazid, rifampicin,

pyrazinamide, and ethambutol has made treatment more challenging [54–56]. The BCG vaccine is not 100% effective against TBM [57]. However, the development of therapeutic vaccines has addressed this limitation. These vaccines are administered to individuals who are already showing symptoms of the disease [58]. In this study, a multi-epitope mRNA therapeutic vaccine was designed using two proteins, *PknD* (*Rv0931c*) and *Rv0986*, from *Mtb* strain H37Rv. Previous research has indicated that *PknD* plays a crucial role in invading the brain endothelium and is a significant microbial factor in CNS diseases [18]. *Rv0986* is part of the three-gene operon *RV0986-88*, which shows strong expression in human blood-brain barrier models during infection and is associated with the virulence and adherence of *Mtb* [19]. Our study reveals that both *PknD* and *Rv0986* exhibit high antigenicity, low sensitization, and non-toxicity, making them promising candidates for developing multi-epitope vaccines. Furthermore, sequence alignment analysis indicates homology between the two proteins, fulfilling the criteria for vaccine design.

SignalP6.0 was utilized to predict the signal peptides of *PknD* and *Rv0986*. Signal peptides, short peptides located at the N-terminal of proteins, are prevalent in both prokaryotes and eukaryotes, influencing protein translation [59, 60]. The unique structural characteristics of signal peptides play a crucial role in the folding and transportation of proteins. Substituting the signal peptide can alter the protein's expression level [61]. Interestingly, our analysis revealed the absence of signal peptides in either egg white, prompting further investigation.

mRNA vaccines induce responses from both CD4+ and CD8+ T cells, activating both the innate and adaptive immune systems in a balanced manner with a specific response to antigens [62]. Helper T lymphocytes (HTL) initiate both humoral and cell-mediated immune responses, while cytotoxic T lymphocytes (CTL) work to halt the spread of viruses by eliminating virus-infected cells and releasing antiviral cytokines. B lymphocytes play a role in humoral immunity, as they are triggered by antigens to produce memory cells and plasma cells that produce various specific antibodies in reaction to antigens [63, 64]. In order to identify appropriate vaccine candidates, we utilized various online tools to predict CTL, HTL, and B cell epitopes(BES) epitopes [65].

Multi-epitope mRNA vaccines have the ability to stimulate an immune response, leading to the production of cellular and humoral immunity [66–68]. In this study, we utilized IEDB, NetCTLpan1.1, and NetMHCIIpan-4.0 to predict CTL and HTL epitopes, while B-cell epitopes were predicted using IEDB and SVMtrip. Specific connectors were used to link antigen epitopes [69]. The mRNA vaccine structure was optimized for translation and stabilization by incorporating various elements such as 5′ m7G cap sequences, a Poly (A) tail, Globin 5'and 3′UTRs flanking the ORF of the mRNA, an adjuvant, the Kozak sequence, a tPA secretion signal sequence, and the MITD sequence. Molecular docking was performed using Xinjiang high frequency alleles to assess the binding of ligands and receptors in vaccine design. Immunological simulations involved three vaccination doses to evaluate the vaccine's capacity to elicit humoral and cellular immune responses. The results demonstrated the effectiveness of the vaccination in pathogen elimination, as indicated by a significant increase in IFN-γ over time post-injection, providing further validation for the accuracy of our vaccine design [70–72].

Escherichia coli was selected as the host organism for expressing the recombinant protein [73]. The online codon optimization tool Jact was employed to optimize the amino acid sequence of the vaccine. Subsequently, upstream and downstream primers were added, with *BamHI* and *XhoI* cleavage sites inserted at the 5' and 3' ends, respectively, to facilitate polymerase chain reaction amplification of the target gene. The vaccine sequence was then cloned into the PVAX1 vector, resulting in a recombinant plasmid of 4098 base pairs. Finally, electron agarose gel electrophoresis was conducted to analyze the target gene, vector, and recombinant plasmid.

TLR-4, a receptor known to be recognized by *Mtb*, has been shown to activate macrophages and dendritic cells, leading to both innate and adaptive immunity [74]. Upon docking the vaccine with TLR-4, we observed a high binding affinity. The stability of the complex was confirmed through the RMSD diagram and further analyzed using molecular dynamics (MD) simulation. In summary, we employed various in silico methods to design an mRNA therapeutic vaccine targeting CNS TB, laying the groundwork for potential future experimental investigations.

## Supporting information

**S1 Table. MHC-I binding prediction results of Rv0986(IEDB).**
(DOCX)

**S2 Table. MHC-I binding prediction results of Rv0986(NetCTLpan-1.1).**
(DOCX)

**S3 Table. MHC-I binding prediction results of PknD(IEDB).**
(DOCX)

**S4 Table. MHC-I binding prediction results of PknD(NetCTLpan-1.1).**
(DOCX)

**S5 Table. MHC-II binding prediction results of Rv0986(IEDB).**
(DOCX)

**S6 Table. MHC-II binding prediction results of Rv0986(NetMHCIIpan version 4.0.**
(DOCX)

**S7 Table. MHC-II binding prediction results of PknD(IEDB).**
(DOCX)

**S8 Table. MHC-II binding prediction results of PknD(NetMHCIIpan version 4.0).**
(DOCX)

**S9 Table. LBEs results of Rv0986 and PknD(SVMtrip).**
(DOCX)

**S1 Fig. Prediction of signal peptides by signal BLAST(PknD).**
(TIF)

**S2 Fig. Prediction of signal peptides by signal BLAST(Rv0986).**
(TIF)

**S3 Fig. Homology comparison by NCBI BlastP.**
(TIF)

## Author Contributions

**Data curation:** Huidong Shi.

**Software:** Huidong Shi, Yuejie Zhu, Kaiyu Shang, Tingting Tian, Zhengwei Yin, Juan Shi, Yueyue He, Jianbing Ding.

**Validation:** Huidong Shi.

**Visualization:** Yuejie Zhu, Kaiyu Shang, Tingting Tian, Zhengwei Yin, Juan Shi, Yueyue He, Jianbing Ding.

Writing – **original draft:** Huidong Shi.

Writing – **review & editing:** Quan Wang, Fengbo Zhang.

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
