## [Decision Letter · Decision Letter 0]

15 May 2024

PONE-D-24-15467Development of innovative multi-epitope mRNA vaccine against central nervous system tuberculosis using in silico approachesPLOS ONE

Dear Dr. Zhang,

Thank you for submitting your manuscript to PLOS ONE. After careful consideration, we feel that it has merit but does not fully meet PLOS ONE’s publication criteria as it currently stands. Therefore, we invite you to submit a revised version of the manuscript that addresses the points raised during the review process.

Please submit your revised manuscript by Jun 29 2024 11:59PM If you will need more time than this to complete your revisions, please reply to this message or contact the journal office at plosone@plos.org. Please include the following items when submitting your revised manuscript:A rebuttal letter that responds to each point raised by the academic editor and reviewer(s). You should upload this letter as a separate file labeled 'Response to Reviewers'.A marked-up copy of your manuscript that highlights changes made to the original version. You should upload this as a separate file labeled 'Revised Manuscript with Track Changes'.An unmarked version of your revised paper without tracked changes. You should upload this as a separate file labeled 'Manuscript'.

We look forward to receiving your revised manuscript.

Kind regards,

Satish Rojekar, Ph.D.

Academic Editor

PLOS ONE

3. Please upload a new copy of Figures 1, 2, 3A, 3B, 6, 7, 8, 9A, 9B, 9C, 12A and 14 as the detail is not clear. Please follow the link for more information: https://blogs.plos.org/plos/2019/06/looking-good-tips-for-creating-your-plos-figures-graphics/" https://blogs.plos.org/plos/2019/06/looking-good-tips-for-creating-your-plos-figures-graphics/

Reviewers' comments:

Reviewer's Responses to Questions

**Comments to the Author**

1. Is the manuscript technically sound, and do the data support the conclusions?

Reviewer #1: Yes

Reviewer #2: Yes

Reviewer #3: Yes

Reviewer #4: No

Reviewer #5: Partly

2. Has the statistical analysis been performed appropriately and rigorously? 

Reviewer #1: Yes

Reviewer #2: N/A

Reviewer #3: Yes

Reviewer #4: N/A

Reviewer #5: Yes

3. Have the authors made all data underlying the findings in their manuscript fully available?

Reviewer #1: Yes

Reviewer #2: Yes

Reviewer #3: Yes

Reviewer #4: Yes

Reviewer #5: Yes

4. Is the manuscript presented in an intelligible fashion and written in standard English?

Reviewer #1: Yes

Reviewer #2: Yes

Reviewer #3: Yes

Reviewer #4: Yes

Reviewer #5: No

5. Review Comments to the Author

Reviewer #1: This insilico approach of developing a multiepitope vaccine for CNS TB is an innovative and thoroughly carried out study. The comprehensive bioinformatics analyses, including sequence homology, antigenicity prediction, and epitope mapping, contribute to a comprehensive understanding of the vaccine candidates' immunogenic potential. Additionally, the incorporation of immunoinformatic tools facilitates the prediction of T-cell and B-cell epitopes, further enhancing the vaccine's efficacy and specificity. Despite the remarkable accomplishments of this study, I have few questions and minor concerns that needs to be addressed.

Please provide clarifications on the following questions:

2. Line 229: Claim “the two proteins are highly homologous, suggesting that they may be derived from the same gene and have similar roles in the immune response”. Did you find any link between the two genes in the previous literatures or perform any additional study to investigate the same other than sequence homology?

3. Did you check similarity between analyzed bacterial proteins and host proteome?

4. Did you by any chance used another software such as Signal BLAST to predict signal peptide?

5. Would you have any other already tested CTL and HTL epitope from previous CNS TB studies / published literature which can be used as comparison with your selected epitope candidates to validate how selected epitope candidates stand out or performing better.

Major concern:

1: Most of the figures are hard to read and see. Image quality is poor and written text is unclear. Please replace with good quality images.

Minor concern:

1. There are various formatting errors like invalid spacing between words in headings and paragraphs. Please check thoroughly.

2. Line 40: Rephrase. Why would TB report 2022 will predict cases in 2021. Prediction happens for future.

3. Line 260: “Docking result between CTLs and HLA-A*02:01 seems confusing. Please rephrase this. Does this mean CTL epitope and HLA-A*02:01?

4. Line 261: Similarly, “Docking result between HTLs and HLA-DRB1*07:01” seems confusing.

5. Section 3.8: Please make a table for MEV properties instead of writing the properties in sentences.

6. Section 9.3: Would it be possible to extrapolate dose and time intervals of injection during simulation of immune response?

7. 3.12 and 3.13 heading both has prediction of secondary structure. Why?

8. Table 4 does not have heading for characteristics measured. No row to define which column represents which characteristics. Include antigenicity values in table 5

Reviewer #2: Following are the comments for the authors to improve on the manuscript -

1. Please use a correct font, font size and text settings for the manuscript

2. Define all the abbreviations in the manuscript for the words where they are used for the first time.

3. Please use the correct style for the citations in the manuscript.

4. The quality and resolution of the images needs to be improved, the aspect ratio for some of the images is incorrect.

5. Please remove the characters from the other languages from the manuscript.

6. please try to reduce the number of images from manuscript, possibly by merging them together.

Reviewer #3: Decision: Minor revision.

General comments:

The authors of this study aimed to develop an innovative mRNA therapeutic vaccine targeting a protein associated with central nervous system tuberculosis, addressing the pressing need for effective treatments for this debilitating condition. Employing in silico techniques such as IEDB, NetCTLpan1.1, NetMHCIIpan-4.0, and SVMtrip, they analyzed the antigen epitopes of PknD and Rv0986. Additionally, CTL, HTL, and B cell epitopes were linked together using AAY, GPGPG, and KK linkers. This research was spurred by the lack of therapeutic options for central nervous system tuberculosis, underscoring its urgency.

The utilization of in silico methods paved the way for designing an mRNA therapeutic vaccine targeting central nervous system tuberculosis, offering a foundation for potential future experimental investigations. The outcomes of these assessments yielded promising observations, indicating the potential efficacy of the developed vaccine.

The authors have presented promising results in the manuscript. However, to optimize its effectiveness for our readers, it would be beneficial to address the following comments for streamlining and enhancement. While the manuscript exhibits good writing and organization, improvements in overall English language usage, particularly in grammar, punctuation, and clarity, would enhance engagement for general readers. Please consider revising the language to maximize effectiveness.

Comments:

1. Line 26: Define ‘BCG’ or provide more descriptive information as it appears first time here in this manuscript. Applicable for all abbreviations throughout the manuscript.

2. Line 33-34: please revise this sentence for effectiveness: ‘The results indicate that the vaccine structure shows promise’.

3. Line 40-41: The tense of the following sentence needs to be revised. If updated and most recent data is available, please provide that. ‘The World Health Organization (WHO) Global TB Report 2022 predicts that 41 10.6 million people will be diagnosed with TB in 2021, with 1.6 million fatalities.’

4. Line 42: Consider defining as this term is used many times in the manuscript ‘Central nervous system tuberculosis’.

5. Line 42: ‘TB’ is not defined, please define it.

6. Line 44: If authors could provide specifics in terms of specific mortality rate from the cited article, that would be helpful for readers.

7. Line 77 to 83: This paragraph has three references cited. This is confusing. Have these previous studies been conducted already? if they are, how they are related to the current study is not clear? please revise the language for clarity. If the intention is to cite for the part of the study, please make it clear.

8. Please revise the whole manuscript for formatting, such as: provide space between the last word in the sentence and the reference parenthesis. Applicable throughout the manuscript.

9. Line 87: link for the database should be before full stop. Please check the formatting throughout the manuscript.

10. 1All figures: Presently, all figures appear distorted and oddly stretched. Please recreate them with appropriate resolution and dimensions to ensure clarity. Additionally, consider increasing the font size on the figures for improved legibility. Some content on certain figures is currently illegible due to distortion. This feedback applies to all figures.

11. All section titles: provide a space between the number and section title, applicable for all section titles.

12. Validation of Models: It is advisable to elaborate on the validation process of the models utilized in this study to assess their reliability. Please address any limitations associated with validating these models to provide insight for readers or researchers interested in utilizing them in the future.

Reviewer #4: The study investigates the innovative epitopes mRNA vaccine by in silico approach. They have utilized bioinformatics tools to justify the epitopes could be the potential candidates for mRNA vaccine. Overall paper makes contribution in finding the novel vaccines for CNS TB. Recommendations for improvements are included in the attachment.

Reviewer #5: In this review, Shi et al, explained a in silico approach for the treatment of CNS Tuberculosis. The authors have explained a detail approach from the selection of the target sequences to codon optimization probabilities to ensure it’s translation in Human host. The authors needs to address some of the queries

6. PLOS authors have the option to publish the peer review history of their article (what does this mean?). If published, this will include your full peer review and any attached files.

Reviewer #1: No

Reviewer #2: No

Reviewer #3: No

Reviewer #4: No

Reviewer #5: **Yes: **Pallapati Anusha Rani

---

## [Author Response · Author response to Decision Letter 0]

21 Jun 2024

Dear editor & reviewers,

Thank you for your kind patience. We have revised our manuscript (Submission ID: PONE-D-24-15467) carefully by following the guidance provided by the editor and reviewers. Here are the point-by-point responses for the editor and reviewers’ comments.

Response to Academic Editor

1.Please ensure that your manuscript meets PLOS ONE's style requirements.

Response:Dear academic editor,thank you for reviewing our research article and providing valuable suggestions for revisions.We have changed the style of the literature according to the requirements of the PLOS ONE journal reference.

2.Please note that PLOS ONE has specific guidelines on code sharing for submissions in which author-generated code underpins the findings in the manuscript. In these cases, we expect all author-generated code to be made available without restrictions upon publication of the work.

Response:Dear academic editor,thank you for reviewing our research article and providing valuable suggestions for revisions. We are very willing to share the code, We have stored laboratory program in a separate protocols.io,generate DOI: dx.doi.org/10.17504/protocols.io.81wgbzr31gpk/v1.; The generated link is: https://www.protocols.io/private/C80855412E6C11EFA80E0A58A9FEAC02, is available for editors and referees to check. We added this section to the Data availability statement. If you need any information from us, please feel free to communicate with us.

3. Please upload a new copy of Figures 1, 2, 3A, 3B, 6, 7, 8, 9A, 9B, 9C, 12A and 14 as the detail is not clear.

Response:Dear academic editor,thank you for reviewing our research article and providing valuable suggestions for revisions.For ease of understanding, we have reuploaded all images as per the magazine's image requirements.

Response to Reviewer1

This insilico approach of developing a multiepitope vaccine for CNS TB is an innovative and thoroughly carried out study. The comprehensive bioinformatics analyses, including sequence homology, antigenicity prediction, and epitope mapping, contribute to a comprehensive understanding of the vaccine candidates' immunogenic potential. Additionally, the incorporation of immunoinformatic tools facilitates the prediction of T-cell and B-cell epitopes, further enhancing the vaccine's efficacy and specificity. Despite the remarkable accomplishments of this study, I have few questions and minor concerns that needs to be addressed.

Response:Dear reviewer,thank you for reviewing our research article and providing valuable suggestions for revisions. We take your feedback seriously and have made corresponding revisions based on your suggestions. Here is our response to your proposed modification suggestions:

1.Line 229: Claim “the two proteins are highly homologous, suggesting that they may be derived from the same gene and have similar roles in the immune response”. Did you find any link between the two genes in the previous literatures or perform any additional study to investigate the same other than sequence homology?

Response:Dear reviewer, thank you for your valuable suggestions.In previous studies, pknD and Rv0986 genes have been shown to be involved in invasion of the central nervous system and are specifically expressed in the central nervous system, missing in lung tissue. In the Uniprot database, both genes were found to be from the H37Rv strain. The references are as follows：

(1).Be NA, Bishai WR, Jain SK. Role of Mycobacterium tuberculosis pknD in the pathogenesis of central nervous system tuberculosis. BMC Microbiol. 2012;12:7. Published 2012 Jan 13. doi:10.1186/1471-2180-12-7；

(2).Be NA, Lamichhane G, Grosset J, et al. Murine model to study the invasion and survival of Mycobacterium tuberculosis in the central nervous system [published correction appears in J Infect Dis. 2009 Jan 15;199(2):290]. J Infect Dis. 2008;198(10):1520-1528. doi:10.1086/592447；

(3).Jain SK, Paul-Satyaseela M, Lamichhane G, Kim KS, Bishai WR. Mycobacterium tuberculosis invasion and traversal across an in vitro human blood-brain barrier as a pathogenic mechanism for central nervous system tuberculosis. J Infect Dis. 2006;193(9):1287-1295. doi:10.1086/502631.

These articles are cited in manuscripts.

2.Did you check similarity between analyzed bacterial proteins and host proteome?

Response:Dear reviewer, thank you for your valuable suggestions.We build a multi-epitope vaccine, and the sequence used to build the vaccine is just a part of the protein. In order to avoid homology between the vaccine and the host, we used the BlastP database to compare homology between the vaccine and Homo sapiens (TaxID: 9606). An E value greater than 0.5 is considered non-homologous and suitable for vaccine construction. The results showed that no significant correlation was found between the two. We have added this section in Line145-149 and Line 324-326.The screenshot of the result is in the supporting information and is called S3(TIF).

3.Did you by any chance used another software such as Signal BLAST to predict signal peptide?

Response:Dear reviewer, thank you for your valuable suggestions.According to your suggestion, we used Signal BLAST to predict the signal peptides of the proteins PknD and Rv0986 again, and the results showed that there were no signal peptides here.The screenshot of the result is in the supporting information and is called S1(TIF), S2 (TIF).

4.Would you have any other already tested CTL and HTL epitope from previous CNS TB studies / published literature which can be used as comparison with your selected epitope candidates to validate how selected epitope candidates stand out or performing better.

Response:Dear reviewer, thank you for your valuable suggestions.The proteins we selected appear specifically in the central nervous system, and there is currently no literature analyzing multiepitope vaccines for CNS TB, which is also the innovation of our paper. However, there are many literature reports on tuberculosis multi-epitope vaccines. Compared with their selected CTL/HTL epitopes, the epitopes we selected all have higher antigenicity, non-allergenic, non-toxic, and good physicochemical properties. The references are as follows:

(1).Andongma BT, Huang Y, Chen F, et al. In silico design of a promiscuous chimeric multi-epitope vaccine against Mycobacterium tuberculosis. Comput Struct Biotechnol J. 2023;21:991-1004. Published 2023 Jan 16. doi:10.1016/j.csbj.2023.01.019

(2).Al Tbeishat H. Novel In Silico mRNA vaccine design exploiting proteins of M. tuberculosis that modulates host immune responses by inducing epigenetic modifications. Sci Rep. 2022;12(1):4645. Published 2022 Mar 17. doi:10.1038/s41598-022-08506-4

(3).Jiang F, Han Y, Liu Y, et al. A comprehensive approach to developing a multi-epitope vaccine against Mycobacterium tuberculosis: from in silico design to in vitro immunization evaluation. Front Immunol. 2023;14:1280299. Published 2023 Nov 2. doi:10.3389/fimmu.2023.1280299

Major concern

1: Most of the figures are hard to read and see. Image quality is poor and written text is unclear. Please replace with good quality images.

Response:Dear reviewer, thank you for your valuable suggestions.We have made changes to the quality and resolution of all images in accordance with the magazine's image requirements.

Minor concern:

1. There are various formatting errors like invalid spacing between words in headings and paragraphs. Please check thoroughly.

Response:Dear reviewer, thank you for your valuable suggestions.We have thoroughly checked the word spacing between the words in headings and paragraph in the manuscript to ensure compliance with the journal requirements.

2.Line 40: Rephrase. Why would TB report 2022 will predict cases in 2021. Prediction happens for future.

Response:Dear reviewer,we are sorry for the tense error due to our negligence. We have rephrase the sentence in line 39-40.

3.Line 260: “Docking result between CTLs and HLA-A*02:01 seems confusing. Please rephrase this. Does this mean CTL epitope and HLA-A*02:01?

Response:Dear reviewer,We are very sorry for the error of expression due to our carelessness. In line 260, we want to express the choice of an ITAPWGIAV epitope in the CTLs to interface with HLA-A*02:01. We have corrected it in the line 302.

4. Line 261: Similarly, “Docking result between HTLs and HLA-DRB1*07:01” seems confusing.

Response:We are very sorry for the error of expression due to our carelessness. In line 261, we want to express that one of the FQFFNLIPTLTVLEN opes in HTLs is selected to interface with HLA-DRB1*07:01. We have corrected it in the line 303-304.

5.Section 3.8: Please make a table for MEV properties instead of writing the properties in sentences.

Response:Dear reviewer, thank you for your valuable suggestions.We have made a table for MEV properties and named it Table 6.

6.Section 3.9: Would it be possible to extrapolate dose and time intervals of injection during simulation of immune response?

Response:Dear reviewer,we have described in section 2.10 three doses of the vaccine for 1 day, 84 days, 168 days, each dose of 1000 units.Based on the simulations, the doses and intervals we injected were reasonable. We have added this sentence to line 349.

7.3.12 and 3.13 heading both has prediction of secondary structure. Why?

Response:Dear reviewer, 3.12 predicts the secondary structure of mRNA in order to ensure the effectiveness of mRNA transcription. 3.13 predicts the secondary structure of the vaccine protein in order to ensure the validity of the translation. For ease of understanding, we have been explained in the 2.12 and 2.13.

8.Table 4 does not have heading for characteristics measured. No row to define which column represents which characteristics. Include antigenicity values in table 5.

Response:Dear reviewer, we are sorry for the problem caused by our oversight. We have added a title to Table4. The antigenicity value is added in Table5. Finally, the antigenicity of MEV we constructed is 0.9510, which has strong antigenicity.

Response to Reviewer2

1.Please use a correct font, font size and text settings for the manuscript

Response:Dear reviewer, thank you for your valuable suggestions.We have used the correct font and font size for the text in the manuscript as required by the journal. Strains and restriction enzyme names are used in italics.

2.Define all the abbreviations in the manuscript for the words where they are used for the first time.

Response:Dear reviewer, thank you for your valuable suggestions.We have defined all the abbreviations used for the first time in the manuscript.

3.Please use the correct style for the citations in the manuscript.

Response:Dear reviewer, thank you for your valuable suggestions.We have changed the style of the literature according to the requirements of the PLOS ONE journal reference.

4. The quality and resolution of the images needs to be improved, the aspect ratio for some of the images is incorrect.

Response:Dear reviewer, thank you for your valuable suggestions.We have made changes to the quality and resolution of all images in accordance with the magazine's image requirements.

5. Please remove the characters from the other languages from the manuscript.

Response:Dear reviewer, thank you for your valuable suggestions.We have checked the language and font in the manuscript to make sure it meets the requirements of the journal.

6.please try to reduce the number of images from manuscript, possibly by merging them together.

Response:Dear reviewer, thank you for your valuable suggestions.We have merged Fig3A,Fig3B into Fig3, Fig9A,B,C into Fig9, Fig11A,B into Fig11, Fig12A,B,C,D into Fig12, Fig13A,B,C into Fig13, Fig14A,B,C into Fig14.

Response to Reviewer3

General comments:

The authors of this study aimed to develop an innovative mRNA therapeutic vaccine targeting a protein associated with central nervous system tuberculosis, addressing the pressing need for effective treatments for this debilitating condition. Employing in silico techniques such as IEDB, NetCTLpan1.1, NetMHCIIpan-4.0, and SVMtrip, they analyzed the antigen epitopes of PknD and Rv0986. Additionally, CTL, HTL, and B cell epitopes were linked together using AAY, GPGPG, and KK linkers. This research was spurred by the lack of therapeutic options for central nervous system tuberculosis, underscoring its urgency.

The utilization of in silico methods paved the way for designing an mRNA therapeutic vaccine targeting central nervous system tuberculosis, offering a foundation for potential future experimental investigations. The outcomes of these assessments yielded promising observations, indicating the potential efficacy of the developed vaccine.

The authors have presented promising results in the manuscript. However, to optimize its effectiveness for our readers, it would be beneficial to address the following comments for streamlining and enhancement. While the manuscript exhibits good writing and organization, improvements in overall English language usage, particularly in grammar, punctuation, and clarity, would enhance engagement for general readers. Please consider revising the language to maximize effectiveness

Response:Dear reviewer,thank you for your full recognition of our manuscript, we have fully considered your comments, made grammatical changes to the full text, checked the use of punctuation, and re-uploaded all images in accordance with the magazine's requirements for image clarity. Here is our response to your comments.

1.Line 26: Define ‘BCG’ or provide more descriptive information as it appears first time here in this manuscript. Applicable for all abbreviations throughout the manuscript.

Response:Dear reviewer, thank you for your valuable suggestions.We have defined BCG and added it to the abbreviations at the end of the manuscript .

2. Line 33-34: please revise this sentence for effectiveness: ‘The results indicate that the vaccine structure shows promise’.

Response:Dear reviewer, thank you for your valuable suggestions.We have rephrased the sentence in lines 32.

3.Line 40-41: The tense of the following sentence needs to be revised. If updated and most recent data is available, please provide that. ‘The World Health Organization (WHO) Global TB Report 2022 predicts that 41 10.6 million people will be diagnosed with TB in 2021, with 1.6 million fatalities.

Response:Dear reviewer, we are sorry for the tense error due to our negligence. We have rephrase the sentence in line 39-40.

4.Line 42: Consider defining as this term is used many times in the manuscript ‘Central nervous system tuberculosis’.

Response:Dear reviewer, thank you for your valuable suggestions.We have defined central nervous system tuberculosis in line 23.

5.Line 42: ‘TB’ is not defined, please define it.

Response:Dear reviewer, thank you for your valuable suggestions.We have defined TB in line 23.

6.Line 44: If authors could provide specifics in terms of specific mortality rate from the cited article, that would be helpful for readers.

Response:Dear reviewer, thank you for your valuable suggestions.We have provided specific details of mortality in line 42-44 and have cited relevant literature.The references are as follows:

(1).Wilkinson RJ, Rohlwink U, Misra UK, et al. Tuberculous meningitis. Nat Rev Neurol. 2017;13(10):581-598. doi:10.1038/nrneurol.2017.120

(2).Mezochow A, Thakur K, Vinnard C. Tuberculous Meningitis in Children and Adults: New Insights for an Ancient Foe. Curr Neurol Neurosci Rep. 2017;17(11):85. Published 2017 Sep 20. doi:10.1007/s11910-017-0796-0

7.Line 77 to 83: This paragraph has three references cited. This is confusing. Have these previous studies been conducted already? if they are, how they are related to the current study is not clear? please revise the language for clarity. If the intention is to cite for the part of the study, please make it clear.

Response:Dear reviewer, thank you for your valuable suggestions.In lines 77-83, we only want to refer to the method part of the literature. The literature is relevant to the design of multi-epitope vaccines, so these methods are also applicable in this study.We have explained this in lines 73-74

8.Please revise the whole manuscript for formatting, such as: pr

---

## [Decision Letter · Decision Letter 1]

15 Jul 2024

Development of innovative multi-epitope mRNA vaccine against central nervous system tuberculosis using in silico approaches

PONE-D-24-15467R1

Dear Dr. Zhang,

We’re pleased to inform you that your manuscript has been judged scientifically suitable for publication and will be formally accepted for publication once it meets all outstanding technical requirements.

Kind regards,

Satish Rojekar, Ph.D.

Academic Editor

PLOS ONE

Additional Editor Comments (optional):

The author must check image quality and formatting, which can be improved and replaced during manuscript proofing.

Reviewers' comments:

Reviewer's Responses to Questions

**Comments to the Author**

1. If the authors have adequately addressed your comments raised in a previous round of review and you feel that this manuscript is now acceptable for publication, you may indicate that here to bypass the “Comments to the Author” section, enter your conflict of interest statement in the “Confidential to Editor” section, and submit your "Accept" recommendation.

Reviewer #1: All comments have been addressed

Reviewer #3: All comments have been addressed

Reviewer #4: All comments have been addressed

Reviewer #5: All comments have been addressed

2. Is the manuscript technically sound, and do the data support the conclusions?

Reviewer #1: Yes

Reviewer #3: Yes

Reviewer #4: Yes

Reviewer #5: Yes

3. Has the statistical analysis been performed appropriately and rigorously? 

Reviewer #1: Yes

Reviewer #3: Yes

Reviewer #4: Yes

Reviewer #5: Yes

4. Have the authors made all data underlying the findings in their manuscript fully available?

Reviewer #1: Yes

Reviewer #3: Yes

Reviewer #4: Yes

Reviewer #5: Yes

5. Is the manuscript presented in an intelligible fashion and written in standard English?

Reviewer #1: Yes

Reviewer #3: Yes

Reviewer #4: Yes

Reviewer #5: Yes

6. Review Comments to the Author

Reviewer #1: The review still need minor revision for formatting errors to enhance the readability of the article. Some of the images are still not clear and needs replacement.

Reviewer #3: Thank you for your prompt and comprehensive responses to my critical questions regarding your manuscript. I appreciate the rigor and thoroughness with which you addressed each point. Your detailed explanations have satisfactorily clarified my concerns, and I am now confident in the validity and robustness of your work.

Based on your responses and the thorough explanations provided, I am pleased to accept your manuscript for publication.

Reviewer #4: I think that the authors have adequately addressed the comments made in the

revised version of the manuscript. Therefore, I have no further comments.

Reviewer #5: In this review, Shi et al, explained a in silico approach for the treatment of CNS Tuberculosis. The authors have explained a detail approach from the selection of the target sequences to codon optimization probabilities to ensure it’s translation in Human host.

Authors addressed all of the major concerns mentioned in the previous report.

7. PLOS authors have the option to publish the peer review history of their article (what does this mean?). If published, this will include your full peer review and any attached files.

Reviewer #1: No

Reviewer #3: No

Reviewer #4: No

Reviewer #5: **Yes: **PALLAPATI ANUSHA RANI

---

## [Editor Report · Acceptance letter]

26 Aug 2024

PONE-D-24-15467R1 

PLOS ONE

Dear Dr. Zhang, 

I'm pleased to inform you that your manuscript has been deemed suitable for publication in PLOS ONE. Congratulations! Your manuscript is now being handed over to our production team.

Kind regards, 

on behalf of

Dr. Satish Rojekar 

Academic Editor

PLOS ONE